# PRETRAINING IN ACTOR-CRITIC REINFORCEMENT LEARNING FOR ROBOT LOCOMOTION

## ABSTRACT

The pretraining-finetuning paradigm has facilitated numerous transformative advancements in artificial intelligence research in recent years. However, in the domain of reinforcement learning (RL) for robot locomotion, individual skills are often learned from scratch despite the high likelihood that some generalizable knowledge is shared across all task-specific policies belonging to the same robot embodiment. This work aims to define a paradigm for pretraining neural network models that encapsulate such knowledge and can subsequently serve as a basis for warm-starting the RL process in classic actor-critic algorithms, such as Proximal Policy Optimization (PPO). We begin with a task-agnostic exploration-based data collection algorithm to gather diverse, dynamic transition data, which is then used to train a Proprioceptive Inverse Dynamics Model (PIDM) through supervised learning. The pretrained weights are then loaded into both the actor and critic networks to warm-start the policy optimization of actual tasks. We systematically validated our proposed method with 9 distinct robot locomotion RL environments comprising 3 different robot embodiments, showing significant benefits of this initialization strategy. Our proposed approach on average improves sample efficiency by 36.9% and task performance by 7.3% compared to random initialization. We further present key ablation studies and empirical analyses that shed light on the mechanisms behind the effectiveness of this method.

## 1 INTRODUCTION

The pretraining-finetuning paradigm has enabled recent major breakthroughs in computer vision (He et al., 2022; Lu et al., 2019) and natural language processing (Devlin et al., 2019), most notably in the case of large language models (Touvron et al., 2023; Achiam et al., 2023). In the domain of robotics, a similar methodology with pre-initialization and fine-tuning has been explored in several works that integrate visual-language model (VLM) backbones for manipulation tasks (Brohan et al.; Black et al., 2024; Team et al., 2024; 2025; Barreiros et al., 2025). However, these works only address the pretraining of the vision or language backbones, which have well-studied benefits and strategies, but do not endow robots with information about embodiment. While these imitation learning-based approaches offer good generalization to different tasks, they suffer from low-frequency execution and are primarily demonstrated on stable platforms and environments, rather than on dynamically unstable robotic platforms or under substantial external disturbances.

In robot locomotion control, reinforcement learning (RL) with Proximal Policy Optimization (PPO) Schulman et al. (2017) has been used to successfully achieve a wide range of robust and agile motions (Hwangbo et al., 2019; Miki et al., 2022; Hoeller et al., 2023; Rudin et al., 2025; Choi et al., 2023; Zhang et al., 2025; Siekmann et al., 2021; Yang et al., 2023). However, skill acquisition is slow and resource-intensive because RL is generally sample-inefficient and each new task is typically learned *tabula rasa*, even within the same embodiment. Looking back at model-based control paradigms (Ferrolho et al., 2023; Sleiman et al., 2021; Bellicoso et al., 2019; Murphy et al., 2012), for a specific robot embodiment, there is knowledge that is sharable across solutions to different tasks, *e.g.*, the joint kinematics and dynamics of the model. Motivated by this, we posit that warm-starting RL training in actor-critic architectures by incorporating such embodiment-aware knowledge into the initial model weights has the potential to improve policy performance and sample efficiency.

Our proposed method consists of three stages: exploration-based data collection, pretraining, and reinforcement learning. We first employ an exploration-based data collection strategy to systematically investigate states most likely to appear in the initial stages of the RL process, where the robot learns fundamental concepts about its embodiment, including limb kinematics, dynamics, and basic stability. With the collected data, we then train an embodiment-aware Proprioceptive Inverse Dynamics Model (PIDM). Finally, by initializing the actor-critic structure with the weights of the PIDM model, we provide the RL process with general knowledge from the initial stumbling stages of the vanilla training process, thus facilitate training. Our pretrained weights do not contain task-specific biases, but let them emerge naturally during RL training, as the entire network is updated in an end-to-end fashion.

There are a large body of works on offline-to-online reinforcement learning (Ball et al., 2023; Hansen et al., 2024; Nakamoto et al., 2023) that aim to bootstrap online RL performance by utilization of a reward-labeled offline dataset. However, our method differ in the way that we aim to provide task-agnostic weights initialization for all possible downstream tasks of that specific embodiment. The unknownness and possible variation of downstream MDPs determine that it is impossible to include task-specific reward signal in the pretraining dataset, thus making the methods that require the target MDP to be fully known and free to explore in advance infeasible. Another line of studies have proposed the development of a skill repertoire for robots (Hoeller et al., 2023) or the pretraining of low-level controllers with fine-grained skills (Peng et al., 2022; 2021) that can be used by high-level controllers. Our perspective distinguish itself by the feature that we do not require a dataset comprising expert-level skills or the retraining of the entire pipeline when adding a new skill. Furthermore, in these works, the final performance heavily relies on the quality of the learned skills and their relevance to the task at the fine-tuning stage, as they can not deviate significantly from behaviors in the original dataset.

In contrast to aforementioned research, this work presents a method for smart network initialization in the context of learning robot locomotion with PPO, which outperforms the commonly used random initialization (He et al., 2015) across various tasks with the same embodiment. Our perspective on the problem is novel as we propose a task-agnostic approach that focuses solely on encapsulating embodiment-specific knowledge across tasks. It does not need reward signal of the task-specific downstream MDPs to be present in the pretraining dataset, and serves as a user-friendly plug-in that does not require modifications to the established paradigm of locomotion learning. We validate this approach with a diverse locomotion skill set and multiple robot embodiments consisting of two quadrupeds and one humanoid. Our approach **improves performance by $7.3\%$ and sample efficiency by $36.9\%$**. The main contributions are:

1. A paradigm of embodiment-specific weight initialization for RL in robot locomotion learning, that improves performance and sample efficiency in the training process.

2. The initialization obtained this way is task-agnostic, *i.e.*, applicable to various downstream Partially Observable Markov Decision Process (POMDP) formulations involving different commands, observations, rewards, curricula and terrains, as long as the same robot embodiment is retained.

3. Extensive empirical validation of our proposed approach with various embodiments and tasks showcases significant improvements in performance and sample efficiency.

## 2 RELATED WORKS

**Pretraining representations in RL** Although RL excels on well-specified tasks, its limited sample efficiency remains a key challenge (Jin et al., 2021), which can be improved with pretraining. Xie et al. (2022) systematically summarized the efforts made to introduce the pretraining paradigm into RL, covering perspectives such as exploration, skill discovery, data coverage maximization, and representation learning. The works most related to this are those that pretrain representations using an unlabeled (reward-free) offline dataset. Schwarzer et al. (2021) employ a combination of latent dynamics modelling and unsupervised goal-conditioned RL to pretrain useful representations that can be later fine-tuned to task-specific rewards. Allen et al. (2021) developed an approach to learn Markovian abstract states by combining inverse model estimation and temporal contrastive learning. Zheng et al. (2025) build a probabilistic model to predict which states an agent will visit in the future using flow matching, but it also necessitates the use of its own RL update algorithm

and thus can not be used with existing prevalent RL algorithms. While all the previously mentioned methods operate on unlabeled datasets as ours does, there are a few fundamental differences. Firstly, we address a more complex set of robotic tasks, incorporating high nonlinearity, noisy observation and massive domain randomization, as well as complex reward structures and environments, and provide a demo supporting its sim-to-real transfer capability. Second, one single pretrained model for an embodiment is shown to be successfully transferred to multiple downstream tasks with various formulations of commands and observation spaces. Finally, instead of trying to exhaustively covering transitions that are possible in the environment during pretraining, we focus on bootstrapping the initial learning steps of downstream RL utilizing similar data given the high-dimensionality of robot locomotion tasks. That said, although driven by a similar aim, we find that none of these works constitute directly comparable baselines to our approach.

**Learning dynamics models via deep learning**   Long et al. (2025) surveyed works on learning dynamics models from physical simulators, highlighting different model architectures and utilization strategies. Lutter & Peters (2023) further categorize such models by their reliance on prior knowledge, their degree of interpretability, and whether they enforce physical properties such as energy conservation. To address the sim-to-sim or sim-to-real gap of a trained policy in simulation, Christiano et al. (2016) propose computing what the simulation expects the resulting next state(s) will be, and then relying on a deep-learned inverse dynamics model to deduce the optimal action. This closely relates to our design of splitting the RL policy into a learned actor part and a pretrained inverse dynamics model. Learning dynamics for legged locomotion is challenging due to high non-linearities, sophisticated contact dynamics, and severe sensor noise from impacts. Levy et al. (2024) propose a semi-structured dynamics model consisting of a known *a priori* Lagrangian equation and an ensemble of learned external torque and noise estimators. Our approach makes no such assumptions, and we remain completely model-free. Xu et al. (2025) trained a neural simulator that is stable and accurate over a thousand simulation steps, utilizing a lightweight GPT-2 (Radford et al., 2019) architecture. In contrast, our architecture is significantly more compact and is also exposed to noise, domain randomization, and partial observability of the environment.

**Cross-task locomotion learning**   AMP (Peng et al., 2021) trains a policy that utilizes behaviors contained in the motion dataset to achieve the task objective, by combining task-rewards with style-rewards specified by an adversarial discriminator. ASE (Peng et al., 2022) pretrains a low-level policy to map latent variables to behaviors depicted in the dataset, and later a task-specific high level policy is trained to specify latent variable for directing the low-level policy to accomplish the task goal. Yang et al. (2020) proposed multi-expert learning architecture (MELA), where they first train a set of experts with distinct skills, and then introduce a gating network which synthesize a weighted combination of experts and is finetuned jointly with the experts, resulting in an adaptive policy. Rudin et al. (2025) distill multiple terrain-specific expert policies into a single foundation policy via the DAgger(Ross et al., 2011) algorithm, which is then finetuned on a broader terrain set, and can be further finetuned on unseen terrain of test. All of them assume access to representations of high-utility skills, either in the form of a motion dataset or a set of expert task-specific policies. Close relation is expected between the task at runtime and behaviors of the experts/dataset, e.g. task at runtime can be solved by a combination of skills in those prior knowledge representations, or task at runtime is some task with domain shift (e.g. locomotion on harder terrain). In our work, we do not require access to expert skills directly related to the runtime task, but are interested in the formulation of universal knowledge for locomotion which can facilitate the learning of a possibly wide range of downstream tasks.

## 3   PRELIMINARIES

Motion control problems are typically represented as Partially Observable Markov Decision Processes (POMDPs), where a policy $\pi : \mathcal{O} \to \mathcal{A}$ directly maps observations $\mathcal{O}$ to actions $\mathcal{A}$ and aims to maximize the cumulative reward. The reward function $R(s_t, a_t, s_{t+1})$ encodes task objectives, where $s_t, s_{t+1} \in \mathcal{S}$ are the current and next state, respectively, and $a_t \in \mathcal{A}$ is the action taken at timestep $t$. Specific to robot motion control tasks, the observation is often the conjunction of command $\mathcal{C}$, proprioception $\mathcal{X}$, exteroception $\mathcal{X}_e$, and last action(s) $\mathcal{A}$. Compositions of these spaces are detailed in Appendix 7.

In RL, a large family of actor-critic algorithms (Konda & Tsitsiklis, 1999) has been widely applied in robotics, among which Proximal Policy Optimization (PPO) (Schulman et al., 2017) is particularly prominent. These constitute an important class of RL algorithms that integrate policy optimization with value function estimation. The actor updates the policy that selects actions, while the critic estimates the value function of the current policy, thereby reducing variance and improving the stability of learning.

Existing works on RL (Lee et al., 2020; Miki et al., 2022; Vollenweider et al., 2022; Arm et al., 2024; Stolle et al., 2024; Portela et al., 2025; Sleiman et al., 2024) often parametrize both the actor and critic networks with a simple Multi-Layer Perceptron (MLP) and initialize its weights randomly (He et al., 2015). Due to the large variety of possible observation configurations, task and command specifications, and the diverse number of layers and input dimensions, pretraining a single model for all downstream tasks becomes impractical. We will address this by providing a modular network architecture and a well-defined pretraining task.

## 4 METHODOLOGIES

### 4.1 PROBLEM FORMULATION

Drawing inspiration from model-based control (Ferrolho et al., 2023; Sleiman et al., 2021; Bellicoso et al., 2019; Murphy et al., 2012), in robotics, the system's target state $s_{t+1}$ is either known or usually easier to derive from the task formulation than the action $a_t$ necessary to get there. This is because $a_t$ is always dependent on the robot's dynamics, which in RL is learned indirectly from experience through the simulator.

**Hypothesis 1** *For robot motion control tasks, a neural-network parameterized policy $\pi$ first formulates the intended target state $s_{t+1}$ and afterwards the action $a_t$ necessary to reach that state.*

We empirically demonstrate evidence supporting this hypothesis in Section 5.2 and propose splitting the vanilla MLP structure into multiple distinct blocks (see Figure 3). One of these blocks is our proposed Proprioceptive Inverse Dynamics Model (PIDM), which we define as a mapping $I(a_t \mid x_{t-K:t+1}, a_{t-K:t-1})$, where $x_t \in o_t$ denotes the proprioception at timestep $t$, $a_t$ denotes the action taken at timestep $t$, and $K$ denotes the length of the history sequence.

### 4.2 OVERVIEW

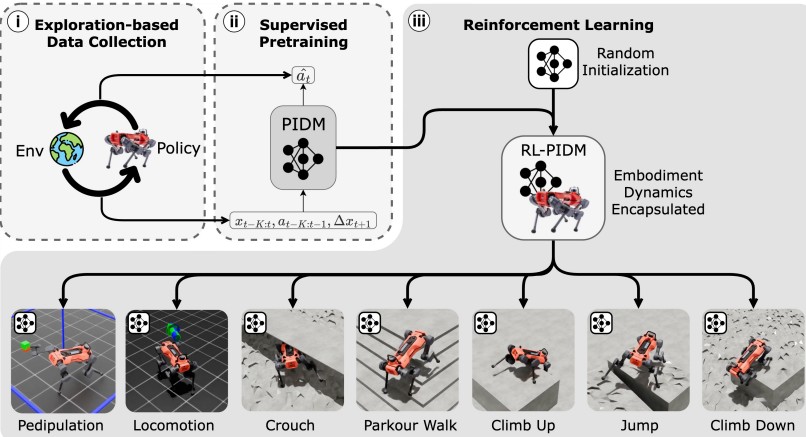

Figure 1: **Method overview:** We (i) collect task-agnostic data using an exploration-driven policy, (ii) to train a Proprioceptive Inverse Dynamics Model (PIDM) to capture embodiment-aware dynamics, and (iii) initialize the actor-critic networks in PPO to warm-start the RL process.

Our overall goal is to pretrain a PIDM model using supervised learning, which can later be integrated into the actor and critic networks of PPO. First, we collect proprioceptive transition data $(x_t, a_t, x_{t+1})$ in a task-agnostic manner from the RL training process of an exploration policy (Sekar

et al., 2020). Important to note is that we solely collect transitions from the early stages of the RL training, rather than from expert policy rollouts for a specific task(s). On the one hand, this design ensures that the method does not rely on prior knowledge of the downstream tasks, nor on access to a (near) expert policy.

On the other hand, the state distribution of randomly initialized policies for different tasks is very similar (see Section 5.3). Therefore, the extracted knowledge should be widely generalizable. By pretraining with this data, the model encapsulates knowledge equivalent to what it would learn in the first iterations of RL (*i.e.*, basic kinematics, dynamics, and stability), enabling it to specialize in learning task-specific skills faster. We integrate the core parts of our pretrained PIDM with randomly initialized outer layers to constitute the actor and critic networks in RL (see Figure 3). Due to the lack of data capturing task-specific dynamics in the pretraining dataset, we allow the PIDM module to be updated in conjunction with the added non-pretrained parts throughout the RL process.

### 4.3 EXPLORATION-BASED DATA COLLECTION

We employ an exploration-based data collection strategy, heavily inspired by previous works (Pathak et al., 2019; Sekar et al., 2020; Curi et al., 2020; Nikolov et al., 2018; Chua et al., 2018), outlined in Figure 2. We use it to obtain data samples that capture the jittery, exploratory behaviors commonly observed in the early stages of RL. In practice, an exploration policy is trained with PPO, where the transitions from the on-policy rollouts are accumulated into a buffer. A probabilistic ensemble of PIDM models is frequently retrained using a bootstrap approach, where data is sampled with replacement from the buffer. The training of the exploration policy is primarily guided by the disagreement in predictions in the ensemble, as a measure of epistemic uncertainty for the PIDM inference. This incentivizes the policy to explore states where the accuracy of the PIDM can be improved with more data. Using the prediction error from a single PIDM model as intrinsic reward may seem probable and easier to implement at first sight. However, we find that its resulting policy is prone to exploring only large-magnitude actions and high-frequency jittering, corresponding to the aleatoric uncertainty of the model. Secondary rewards added include a minimal set of regularizing rewards to constrain unwanted behaviors (e.g., high action rates, torques, or joint velocities) that are common to any task, as well as a term that rewards foot-air-time to encourage interaction with the terrain. During data collection, we employ standard domain randomization techniques for RL training (Miki et al., 2022; Lee et al., 2020; Kaidanov et al., 2024), such as varying the robot link masses, the friction coefficients, and applying random forces.

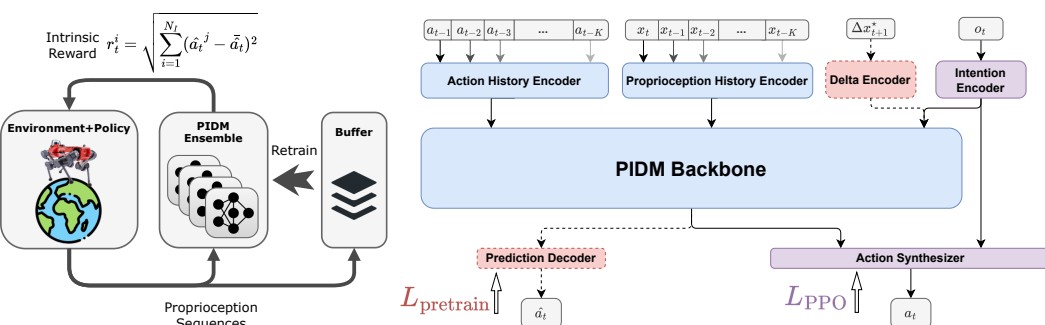

Figure 2: Diagram for exploration-based data collection pipeline, showcasing how the simulation collects data, and is guided by the ensemble of PIDM models that are periodically retrained using the buffered data.

Figure 3: Proprioceptive Inverse Dynamics Model (PIDM) architecture and its integration into the actor network. During pretraining of the PIDM, the dashed red parts of the network are included. However, when integrating into the actor-critic structure, those are removed and replaced by the encoder and decoder in purple.

### 4.4 PRETRAINING THE PROPRIOCEPTIVE INVERSE DYNAMICS MODEL

We parameterize the PIDM with an MLP-based modular architecture, as shown in Figure 3. The model takes as input a history of actions $a_{t-K:t-1}$ and proprioceptive observations $x_{t-K:t+1}$ of length $K$. Both are passed through a dual-layer MLP encoder before being fed into the *PIDM*

*backbone*, which is a 4-layer MLP. During pretraining, we give the model a desired delta-state $\Delta x^*_{t+1}$ to achieve in the next time step. We then use an $L1$ loss to supervise the PIDM to output the required action $a_t$ to reach the target future state $x^*_{t+1}$. The pretraining dataset is also augmented with symmetry transformations, as defined by Mittal et al. (2024) or Byun & Perrault (2024), and observation noise to improve robustness and increase sample diversity (see Appendix A.4).

The necessity of including a history of proprioception for PIDM is mainly due to the absense of terrain information and contact state in proprioception, and the presence of noise and domain randomization techniques in the training process (during both exploration-based data collection and training of task-specific policies). Therefore, it would be inappropriate to fit the PIDM with only one single frame of current proprioceptive state, due to the fact that one certain proprioception can be observed in a range of actual full states in the POMDP. The action and proprioception histories can provide indirect observability of contact states, the domain randomization variables during training (*e.g.*, mass and friction randomization), and of random forces being applied to the robot (Ji et al., 2022; Portela et al., 2025). This knowledge is crucial for mastering the system's dynamics. Meanwhile it is important to note that the PIDM model does not have access to privileged information.

### 4.5 WARM-STARTING REINFORCEMENT LEARNING

**Integrating PIDM into actor-critic networks:** The pretrained PIDM is integrated into both the actor and critic networks. As shown in Figure 3, for the actor, we first remove the *Delta Encoder* and substitute it with a randomly initialized *Intention Encoder* that processes the complete task-specific observation. The *Intention Encoder* now only needs to learn an embedding-based representation of the task-specific delta target state $\Delta x^*_{t+1}$, which can be preprocessed by the pretrained *PIDM Backbone*. Meanwhile, the original output-layer (*Prediction Decoder*) is removed, and the concatenated outputs of the *PIDM Backbone* and *Intention Encoder* are passed in to a randomly initialized *Action Synthesizer* that synthesizes the final action $a_t$. PIDM is used in the critic via an almost identical architecture, with the only difference that the *Action Synthesizer* in the actor is replaced with a *Value Synthesizer* that outputs a scalar value estimation optimized with MSE loss.

The addition of the *Intention Encoder* is necessary to ensure dimension compatibility and enable the training to steer the pretrained module. The task-specific observation $o_t$ can be anything and is wholly independent of our proposed approach. We also empirically discovered that the inclusion of the randomly initialized *Action Synthesizer* is crucial for stabilizing the training by ensuring that the action distribution at the initial stage of RL is similar to that of the case with a randomly initialized vanilla MLP. More specifically, the random initialization of the *Action Synthesizer* ensures near unit-Gaussian action distribution at the beginning, thus avoiding extreme actions that would incur significantly more failures or penalties. Moreover, a final advantage is that, in the event the PIDM is not beneficial for the task, there is a bypass pathway in the structure that facilitates an easy fallback to a classic randomly initialized MLP.

**Intact RL setup:** Except for the architectures of the actor and critic networks and the way the weights are initialized, our method does not require any modifications to either the POMDP (reward, curriculum design, observations, actions, and terminations) or to the PPO update rules, hyperparameters. The task-dependent *Intention Encoder* and *Action Synthesizer* can adapt to any configuration and dimension of the input and output. Therefore, the feasibility of handling arbitrary tasks is not limited. Every parameter in the pretrained PIDM remains trainable during the RL process. In this way, we allow task-specific dynamics to be learned during policy optimization, which eases the burden of attempting to exhaustively cover all possible transitions in the pretraining dataset.

## 5 EXPERIMENTS

### 5.1 REINFORCEMENT LEARNING TASKS

We test our method on 9 RL environments with 3 distinct embodiments: a) 2 blind tasks (velocity-tracking locomotion (Rudin et al., 2022) and pedipulation (Arm et al., 2024)) and 5 perceptive tasks (parkour walk, climb up, climb down, crouch, and jump (Hoeller et al., 2023)) with ANYmal-D (Hutter et al., 2016), b) velocity-tracking locomotion task with Unitree Go1 quadrupedal robot (default implementation in Mittal et al. (2025)), and c) velocity-tracking locomotion task with Unitree

G1 humanoid robot (default implementation in Mittal et al. (2025)). All training is performed in Isaac Lab (Mittal et al., 2025).

Despite the diverse rewards, curricula and hyperparameters involved in the original implementations in aforementioned works, the network architectures used are very similar: the actor and critic networks are both 4-layer MLPs. The compactness of the architectures can be attributed to the fact that the trained policy networks are expected to be deployed on real mobile hardware and be reactive at a high frequency (typically $50 \sim 200$ Hz). Our proposed architecture has approximately $4\times$ the number of parameters due to the inclusion of state history and the need to cover a larger initial state space in pretraining, compared to task-specific policies that can immediately hyper-specialize.

## 5.2 Dynamics Knowledge Probing in Vanilla RL Policy Networks

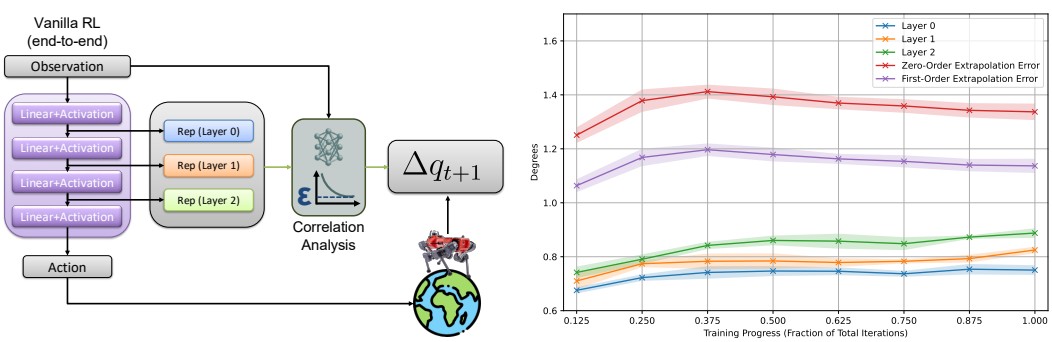

(a) Experiment setup diagram.

(b) Errors of dynamics prediction based on representations from different layers, *Pedipulation* task.

Figure 4: Experiment to probe dynamics knowledge in vanilla policy networks. We analyze the correlation between intermediate representations between layers and the future joint state $q_{t+1}$. The zero-order extrapolation in (b) is a reference of the accuracy of always predicting $q_t = q_{t+1}$. Shaded areas indicate the standard deviation over 5 RL runs.

As a means of empirically studying our initial Hypothesis 1, we examine policy network model checkpoints from some task-specific RL process. For each checkpoint, we collect a number of observation-action pairs from the rollout distribution of the policy corresponding to that checkpoint. We then execute the mean action and record the resulting change in joint angles $\Delta q_{t+1}$. Meanwhile, we collect the intermediate representations from all three hidden layers, as illustrated in Figure 4a.

We investigate how well the network understands at a specific layer what the consequences of its action will be by fitting a lightweight MLP to regress $\Delta q_{t+1}$ based on the tuple consisting of raw observation and intermediate representation corresponding to that layer. The lower final prediction error of a certain configuration indicate a better understanding.

Results of *Pedipulation* task is shown in Figure 4b, and results of *Locomotion* task is shown in Figure 8. In the vertical direction, the correlation between the future state diminishes as we progress deeper into the network. This highlights the analogy of trained vanilla MLP policy networks to classic control from Section 4.1, where the model first forms an intent on the target state and subsequently computes the inverse dynamics to determine the required action. For more details on the experiment, we refer the readers to Appendix A.2.

## 5.3 Pretraining the Proprioceptive Inverse Dynamics Model

In this subsection, we describe how to obtain a pretrained PIDM model and analyze both the dataset distributions and the model's accuracy, using ANYmal D as an example. We first analyze the quality of the data collected using the exploration-based strategy outlined in Section 4.3. In addition to the previously mentioned standard data augmentations (*e.g.*, mass randomization, random noise, symmetry), we collect data on either or both flat and basic rough terrain generated with Perlin noise (Miki et al., 2022; Lee et al., 2020). In Figure 5a we plot samples of ANYmal D from the flat-terrain environments along with samples from the learning process of *Pedipulation* and *Locomotion*

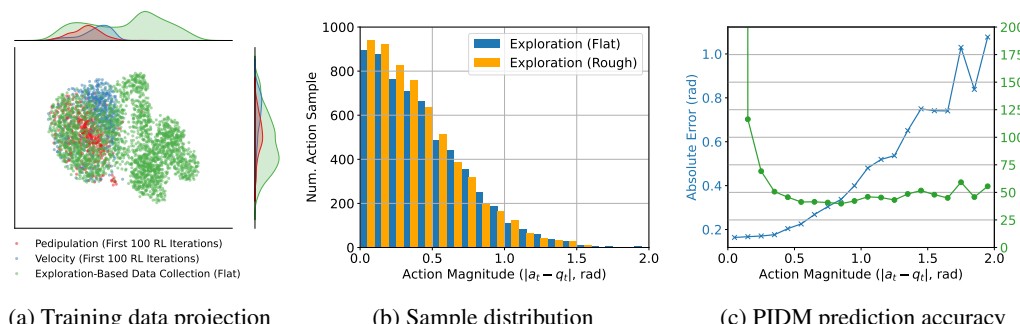

(a) Training data projection     (b) Sample distribution     (c) PIDM prediction accuracy

Figure 5: **PIDM training and dataset analysis of ANYmal D:** For the pretraining dataset we visualize its (a) coverage (green) compared to the initial exploration stages in RL (red and blue) using an UMAP projection and (b) the sample distribution of absolute action magnitudes $|a_t - q_t|$ over different terrains. Finally, in (c) we show the resulting PIDM accuracy across the entire action range as absolute joint errors $|\hat{q}_{t+1} - q_{t+1}|$ and also normalized by the action magnitude.

tasks, which are trained solely on flat terrain. Using UMAP (McInnes et al., 2018), we project the proprioceptive observations $x$ for our collected dataset into 2D, and the observations from the first 100 iterations of RL training for pedipulation and locomotion. We can thus validate that we obtain good coverage of, and beyond, the initial stages of the RL training process, which aligns with the goals outlined in Section 4.2.

The PIDM is pretrained as described in Section 4.4. For each embodiment, we use a total of 5∼7 million samples for training and a similarly sized, disjoint validation set. For plotting purpose only, we randomly select $1,000$ validation samples and consider only the $12$ joint angles $q$ of ANYmal D. Figure 5b shows the distribution of the action magnitude, *i.e.* the magnitude in radians of the commanded changes in joint angles. Figure 5c shows the final prediction accuracy of the trained PIDM. We show both the absolute error and the normalized error, which is the error expressed as a fraction of the action magnitude. It achieves a normalized error of around $40\% \sim 50\%$, with a minimum error of $\sim 0.1$ radians for small actions.

This indicates the considerable difficulty in training a PIDM with high accuracy, which we attribute to the vast transition space, partial observability, and the lack of inductive bias in MLPs. For a detailed discussion, see Appendix A.4. While accurate data-driven modeling of robot locomotion may be achievable with models that are many orders of magnitude larger (Xu et al., 2025) than those used here, actor and critic networks in motion-policy learning have traditionally been extremely lightweight. Moreover, large models are known to make reinforcement learning substantially more challenging (Ota et al., 2021; Li et al., 2023). As a result, significantly increasing model capacity raises concerns about whether existing methods can still be applied without modification. For these reasons, we choose to keep the model size to millions of parameters which is much closer to that of the vanilla MLPs used in prior works. Although they may not seem too accurate for an inverse-dynamics solving, we will demonstrate in the following subsection that a pretrained module of such accuracy can already significantly enhance RL training. We further present a study of relation between error level of PIDM and RL performance in Appendix A.8 to verify that a trend of positive correlation between the accuracy of pretrained PIDM and resulted gain in RL performance can be observed.

### 5.4 QUANTITATIVE EXPERIMENTS

In each experiment, we compare three methods: (i) the vanilla 4-layer MLP, (ii) our PIDM architecture with randomly initialized weights, and (iii) our PIDM architecture with pretrained weights. The utility of our method (*i.e.* using pretrained weights) is indicated by the comparison between (ii) and (iii). The performance of the vanilla 4-layer MLP is included only as a reference. Results are averaged over five runs with different random seeds, except for some individual cases mentioned in Appendix Table 8. To note is that we did *not* tune the learning parameters (learning rate, entropy coefficient, etc.) of the tasks, which were chosen for optimal performance of the original vanilla

| Metric | | Method | ANYmal D | | | | | | | Go1 | G1 | Avg. |
|---|---|---|---|---|---|---|---|---|---|---|---|---|
| | | | Loco-motion | Pedipu-lation | Parkour Walk | Climb Up | Climb Down | Crouch | Jump | Loco-motion | Loco-motion | |
| Final perf. increase | $(\%, \uparrow)$ | Vanilla MLP | +0.5 | +0.2 | -0.8 | 0.0 | +6.5 | +1.8 | +11.1 | +0.6 | -0.2 | +2.2 |
| | | PIDM (Pretrained) | +10.1 | +6.3 | +0.7 | 0.0 | +27.7 | +1.8 | +5.9 | +3.6 | +10.0 | +7.3 |
| Num. iters. to converge | $(\%, \downarrow)$ | Vanilla MLP | -28.7 | +5.0 | -18.7 | +22.5 | -11.4 | -29.5 | -53.0 | +2.2 | -46.7 | -17.6 |
| | | PIDM (Pretrained) | -33.1 | -42.0 | -35.3 | -20.6 | -43.2 | -57.3 | -49.3 | -17.7 | -34.0 | -36.9 |

Table 1: Increase in performance (based on reward/curriculum progress) and sample efficiency (number of iterations required to reach 90% of the maximum performance). Percentage values are *w.r.t.* a randomly initialized PIDM model. Values are averaged across five runs with different starting seeds. We also report the performance of the 4-layer vanilla MLP for reference.

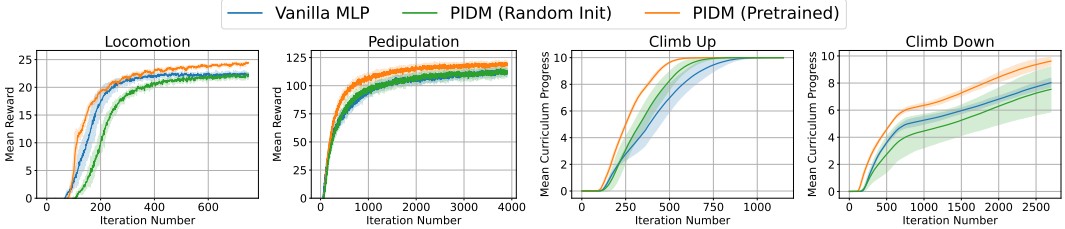

Figure 6: Evolution of the main performance metric during training for *Locomotion*, *Pedipulation*, *Climb Up* and *Climb Down* tasks with ANYmal D. The shaded areas denote standard deviations across five seeds.

MLP. We merely used our architecture as a drop-in replacement. Therefore, it is possible that the performance of the proposed method can be further improved with additional tuning of these hyperparameters, which would have happened if the problem design had used our proposed architecture as a starting point.

We introduce two metrics to quantify the amount of difference in RL performance:

- **Final performance increase** expresses the percentage of change in the main performance indicator of each method compared to that of the randomly initialized PIDM baseline.
- **Number of iterations to converge** is a measure of sample efficiency. This term represents the percentage of change in the number of iterations required to reach 90% of the final performance of the *PIDM (Random Init)* baseline in the main performance indicator.

The selection of the main performance indicator varies across tasks (for details, see Appendix A.6). The results across all nine tasks are presented in Table 1. For some tasks, we also plot the evolution of the main performance indicator during training in Figure 10. The PIDM architecture with random weight initialization, *PIDM (Random Init)*, generally lags behind the vanilla MLPs due to a larger model size and input dimension (inclusion of history). However, with the proposed pretraining strategy, *PIDM (Pretrained)* not only consistently outperforms *PIDM (Random Init)* in all metrics, but also significantly surpasses the performance of the vanilla MLP in 7 out of 9 tasks. Matching the performance of the MLP with our PIDM architecture is a secondary goal that could be potentially achieved by exhaustive tuning of the model and proprioceptive input. The key takeaway is the comparison between the randomly initialized and pretrained architectures. Compared with the vanilla MLP, *PIDM (Pretrained)* demonstrates an average improvement of 5.0% on final performance, and 18.8% on sample efficiency. When compared with *PIDM (Random Init)*, the proposed *PIDM (Pretrained)* showcases an improvement of 7.3% on final performance, and enhances sample efficiency by a margin of 36.9%. We also note that despite the PIDM never having experienced the complex terrains used in the parkour task (see Figure 1), it quickly adapts to the new task-specific dynamics during RL training.

## 5.5 ABLATIONS

We also perform 2 ablations with *Climb Up* and *Climb Down* tasks of ANYmal D, to motivate some of our design choices. First, in Table 2 we analyze the initialization strategy of choosing to pretrain

| Metric | Method | Anymal D Climb Up | Climb Down |
|---|---|---|---|
| Final perf. increase (%, ↑) | PIDM (Pretrained Actor Only) | 0.0 | +19.0 |
|  | PIDM (Pretrained Critic Only) | 0.0 | +17.8 |
|  | PIDM (Pretrained Both) | 0.0 | +27.7 |
| Num. iters. to converge (%, ↓) | PIDM (Pretrained Actor Only) | -11.0 | -37.9 |
|  | PIDM (Pretrained Critic Only) | +18.4 | -28.2 |
|  | PIDM (Pretrained Both) | -20.6 | -43.2 |

Table 2: Ablation on using pretrained weights to initialize either the actor, critic, or both. Results are in comparison to a fully randomly initialized PIDM architecture.

| Metric | Method | Anymal D Climb Up | Climb Down |
|---|---|---|---|
| Final perf. increase (%, ↑) | PIDM (Pedipulation Data) | 0.0 | +24.9 |
|  | PIDM (Exploration Data) | 0.0 | +27.7 |
| Num. iters. to converge (%, ↓) | PIDM (Pedipulation Data) | -10.1 | -40.0 |
|  | PIDM (Exploration Data) | -20.6 | -43.2 |

Table 3: Ablation on data source for pretraining (exploration data versus data from initial RL stages on pedipulation).

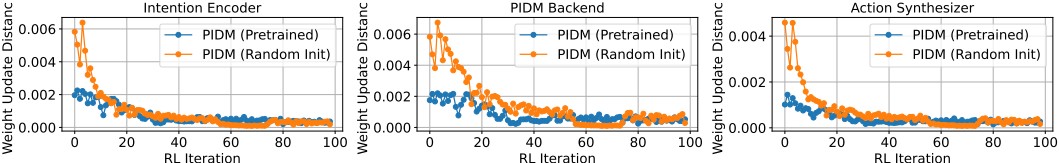

Figure 7: Network weight update magnitude comparison in the PIDM structured actor network during ANYmal D pedipulation RL training. In each submodule, the update of each linear layer weight is indicated by the mean absolute change per parameter, which is then averaged over all layers.

either or both of the actor and critic networks. We note that only pretraining either the actor or critic generally still improves mean performance, but at the cost of increased instability, underscored by larger variation across runs (see Figure 12 in the Appendix). Second, we also ablate the data source used to pretrain the PIDM. We compare using exploration-based data versus samples from the initial stages of RL training a policy (in this case, pedipulation, as shown in Table 3. Notably, both datasets significantly outperform random initialization. Our approach provides an extra margin of improvement and theoretically adapts better to downstream tasks without overcomplicating the pipeline, as pretraining is performed only once.

## 5.6 Weight Update Magnitude

We compare the network weight update magnitudes between *PIDM (Pretrained)* and *PIDM (Random Init)* during the first 100 iterations of RL in Figure 7. We find that not only does the model exhibit smaller updates per iteration in the pretrained *PIDM backbone*, but this also results in smaller updates in the randomly initialized upstream *Intention Encoder* and downstream *Action Synthesizer*. This finding suggests that our pretrained weights lie closer to the desired local minimum and is an indicator that the optimization process can properly leverage this fact. For more examples, see Figure 15 in Appendix.

## 6 Conclusion

To summarize, we have presented a method for warm-starting the RL training process in actor-critic algorithms, targeted for robotic motion control. Our proposed approach leverages a network architecture based on a Proprioceptive Inverse Dynamics Model (PIDM) that is pretrained using exploration-based data from a specific robot embodiment. Our modular architecture functions as a drop-in replacement, without hyperparameter tuning, for any task on the pretrained robot embodiment. We demonstrate on 9 diverse RL environments with 2 quadrupedal robots and 1 humanoid robot that we can **improve the final performance by 7.3%, and enhance sample efficiency by 36.9%**. We also provide ablation studies to motivate our design choices and extensive empirical insights into the inner workings of our method. Future work can focus on optimizing model design to reduce the network size further and incorporating network architectures that are more adept at working with time-series data.

## REPRODUCIBILITY STATEMENT

For reproducibility of our approach, we provide extensive implementation details in the Appendix. Additionally, we have included the source code as part of the submission and intend to open-source it as an extension for IsaacLab after publication, for the benefit of the robotics learning community.

## ETHICS STATEMENT

Our method deals with Reinforcement Learning in simulation. While we do not foresee direct ethics concerns, we acknowledge that our contribution targets advances in both robotics and learning. By promoting GPU-intensive learning algorithms that also require realistic simulation environments and making them more accessible, we contribute to $CO_2$ emissions and the global climate crisis. In parallel, by contributing to advancements in robotics, we facilitate access to more advanced platforms that have the potential to harm humanity. While robots can take over dangerous or monotonous jobs, automation also contributes to jobs lost and economic change. Even more grim is the prospect of using robots for warfare, where motion control and robustness across various terrains for legged platforms are especially crucial factors in enabling such technologies. We promote the responsible use of our work, and hope that it will not be used for harm.

## LARGE LANGUAGE MODEL USE

Large language models (ChatGPT, Gemini) and other writing aids (*e.g.,* Grammarly) are solely used to polish and correct writing in individual sentences, and not to generate entire sections of text.

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

# A APPENDIX

## A.1 PPO ALGORITHM

We use the *RSL_RL* (Schwarke et al., 2025) implementation of PPO with adaptive learning rate and symmetry augmentation in RL (Mittal et al., 2024) (only for parkour tasks). The pseudo code is shown in Algorithm 1.

Notably, on top of the common understanding of PPO, this version has some additional implementation details:

- For general POMDP we are dealing with in robot motion control, the state $s_t$ is approximated by observation $o_t$.
- Policy $\pi_\theta$ is a diagonal Gaussian distribution $\mathcal{N}\big(\mu_{\theta_\mu}(s_t), \sigma_{\theta_\sigma}\big)$, where $\mu_{\theta_\mu}$ is a neural network parameterized by $\theta_\mu$ and produces action mean of each action dimension, and $\theta_\sigma$ is a set of trainable parameters that control the standard deviation of every action dimension.
- All parameters are optimized by Adam (Kingma & Ba, 2014) optimizer, whose learning rate is dynamically adjusted according to the Kullback–Leibler divergence calculated with each minibatch (Algorithm 1 line 17~20).
- When symmetry augmentation is active, new samples produced by symmetry augmentation are added to every minibatch, and the update rule is adapted accordingly. For more details, we refer the readers to Mittal et al. (2024).

## A.2 DYNAMICS KNOWLEDGE PROBING DETAILS

The experiments were performed in *Pedipulation* (Figure 4b) and *Locomotion* (Figure 8) tasks, each with 5 different seeds. Algorithm pseudocode is shown in Algorithm 2. Representation extraction functions $\{f_j\}_{j=0}^{N-1}$ denotes the functions that extract intermediate representation from layer $j$ of one vanilla policy network, parameterized by the parameters of that policy network.

**Collection by deterministic policy on state distribution tied to the stochastic policy** According to the policy parameterization formulation described in Section A.1, the parameters of policy network $\theta_\mu$ does not describe action uncertainty, is thus not accountable for the consequence of an action sampled stochastically from the distribution $\pi_\theta(o_t) = \mathcal{N}\big(\mu_{\theta_\mu}(s_t), \sigma_{\theta_\sigma}\big)$. Therefore, the action-consequence pair needs to be collected by the deterministic policy $a_t \leftarrow \mu_{\theta_\mu}(o_t)$. However, the dynamics knowledge should be assessed on the distribution of the stochastic policy $\pi_\theta(o_t)$. In practice we run sufficient steps with the stochastic policy $\pi_\theta(o_t)$ to make sure the states distribution fit $p_{\pi_\theta}(o_t)$, and then execute mean action from deterministic policy by only one timestep and collect the transition data into dataset. We repeat this preparation-collection loop until we have enough data.

**Lightweight MLP formulation** The correlation analysis is performed by fitting a lightweight MLP to regress the change in joint angles from the concatenation of some internal representation and raw observations. The choice to concatenate raw observation to every intermediate representation is to ensure that, the input to the lightweight MLP contains full information about current state at all times. This makes sure that the differences between the final prediction errors are purely resulted from the understanding of dynamics of intermediate representation, not from how much it contains current state information.

**Limited computation** The computation of fitting the lightweight MLP is restricted in two ways: 1) limited width and depth of MLP, and 2) limited optimizer steps. This is due to the consideration that if the size of the MLP and the computation are more than sufficient with respect to this task and dataset, the network is then able to overfit the training set regardless of how strong the correlation is, and thus produces low error with whichever input configuration. Therefore, we intentionally use a lightweight MLP and a small amount of optimizer steps in all fitting runs. In practice, the trainable lightweight MLP takes the form of a 2-layer MLP, with the only hidden layer's dimension being 64 and activation function being ELU. The lightweight MLP is optimized by Adam with a learning rate of 0.001 for 20 epochs on a training set of 11700 samples.

**Algorithm 1** Proximal Policy Optimization (PPO) — RSL_RL Version with adaptive learning rate

**Require:** parallel environments $\mathcal{E}$
**Require:** initial policy (actor) network parameters $\theta_0$, value (critic) network parameters $\phi_0$
**Require:** clip param $\epsilon$, discount $\gamma$, $\lambda$, ComputeGAE, OptimizerStep, ComputeSymmetryLoss
**Require:** simulation steps per iteration $T$, learning epochs per update $K$, minibatch number $M$
**Require:** initial learning rates $\alpha_0$, value loss coef $c_{\text{vf}}$, entropy coef $c_{\text{ent}}$
**Require:** max grad norm $g_{\max}$, desired KL per optimizer step $\delta_{\text{KL}}$, learning rate adjustment ratio $\eta_\alpha$, max iterations $N$

1: Initialize $\mathcal{E}$
2: $\theta \leftarrow \theta_0,\ \phi \leftarrow \phi_0,\ \alpha \leftarrow \alpha_0$
3: **for** iter $= 0, \ldots, N-1$ **do**
4:     **for** T $= 0, \ldots, K-1$ **do**
5:         Sample an action $a_t$ from the action distribution
        $a_t \sim \pi_\theta(a_t|s_t) = \mathcal{N}\big(\mu_{\theta_\mu}(s_t), \sigma_{\theta_\sigma}\big)$
6:         Step the environments $(s_{t+1}, r_t, \text{done}_t) \leftarrow \mathcal{E}.\text{step}(a_t)$
7:     **end for**
8:     Collect transitions into buffer
        $\mathcal{D} \leftarrow \{(s_t, a_t, r_t, \text{done}_t, \log \pi_\theta(a_t|s_t), V_\phi(s_t))\}_{t=0}^{T-1}$
9:     Let $V_\phi(s_T)$ be bootstrap value (if step $T$ is terminal set to 0, else evaluate $V_\phi$).
10:    Compute advantages and returns:
        $\{\hat{A}_t\}, \{R_t\} \leftarrow \text{COMPUTEGAE}(\{r_t\}, \{V_\phi(s_t)\}, V_\phi(s_T), \gamma, \lambda)$
11:    Normalize advantages: $\hat{A}_t \leftarrow \dfrac{\hat{A}_t - \overline{\hat{A}}}{\text{std}(\hat{A}) + 10^{-8}}$
12:    $\theta_{\text{old}} \leftarrow \theta$
13:    **for** epoch $= 1, \ldots, K$ **do**
14:         Shuffle $\mathcal{D}$ and split into $M$ minibatches $\mathcal{B}$
15:         **for** each minibatch $\mathcal{B}$ **do**
16:             For every sample $(s_t, a_t, R_t, \hat{A}_t, \log \pi_{\theta_{\text{old}}}(a_t|s_t)) \in \mathcal{B}$ compute:
    $\log \pi_\theta(a_t|s_t)$, $V_\phi(s_t)$ and entropy $\mathcal{H}(\pi_\theta(\cdot|s_t))$
17:             $D_{KL} = \frac{1}{|\mathcal{B}|} \sum_{s_t \in B} \log\left(\frac{\sigma_{\theta_\sigma}}{\sigma_{\theta_\sigma^{\text{old}}} + 10^{-5}}\right) + \frac{\sigma_{\theta_\sigma^{\text{old}}}{}^2 + (\mu_{\theta_\mu^{\text{old}}}(s_t) - \mu_{\theta_\mu}(s_t))^2}{2\sigma_{\theta_\sigma}{}^2} - \frac{1}{2}$
18:             **if** $D_{KL} > 2 \cdot \delta_{\text{KL}}$ **then** $\alpha \leftarrow \max(\alpha/\eta_\alpha, 1 \times 10^{-5})$
19:             **else if** $D_{KL} < 0.5 \cdot \delta_{\text{KL}}$ **then** $\alpha \leftarrow \min(\alpha \cdot \eta_\alpha, 1 \times 10^{-2})$
20:             **end if**
21:             ratio: $r_t(\theta) = \exp\big(\log \pi_\theta(a_t|s_t) - \log \pi_{\theta_{\text{old}}}(a_t|s_t)\big)$
22:             clipped surrogate: $s_1 = r_t(\theta)\hat{A}_t, \quad s_2 = \text{clip}(r_t(\theta), 1-\epsilon, 1+\epsilon)\hat{A}_t$
23:             policy loss (to minimize): $\mathcal{L}^{\text{CLIP}} = -\frac{1}{|\mathcal{B}|} \sum_t \min(s_1, s_2)$
24:             value loss: $\mathcal{L}^{\text{VF}} = \frac{1}{|\mathcal{B}|} \sum_t (V_\phi(s_t) - R_t)^2$
25:             entropy bonus: $\mathcal{S} = \frac{1}{|\mathcal{B}|} \sum_t \mathcal{H}(\pi_\theta(\cdot|s_t))$
26:             full loss: $\mathcal{L}(\theta, \phi) = \mathcal{L}^{\text{CLIP}} + c_{\text{vf}}\mathcal{L}^{\text{VF}} - c_{\text{ent}}\mathcal{S}$
27:             **if** use symmetry loss **then** $\mathcal{L}(\theta, \phi) \leftarrow \mathcal{L}(\theta, \phi) + \text{ComputeSymmetryLoss}(\mathcal{B}, \theta)$
28:             **end if**
29:             Compute gradients $\nabla_\theta \mathcal{L}$, $\nabla_\phi \mathcal{L}$
30:             Clip gradients: $\|\nabla\| \leftarrow \min\big(1, g_{\max}/\|\nabla\|\big)\nabla$
31:             Update parameters: $\theta, \phi \leftarrow \text{OptimizerStep}(\theta, \phi, \nabla\mathcal{L}, \alpha)$
32:         **end for**
33:     **end for**
34: **end for**

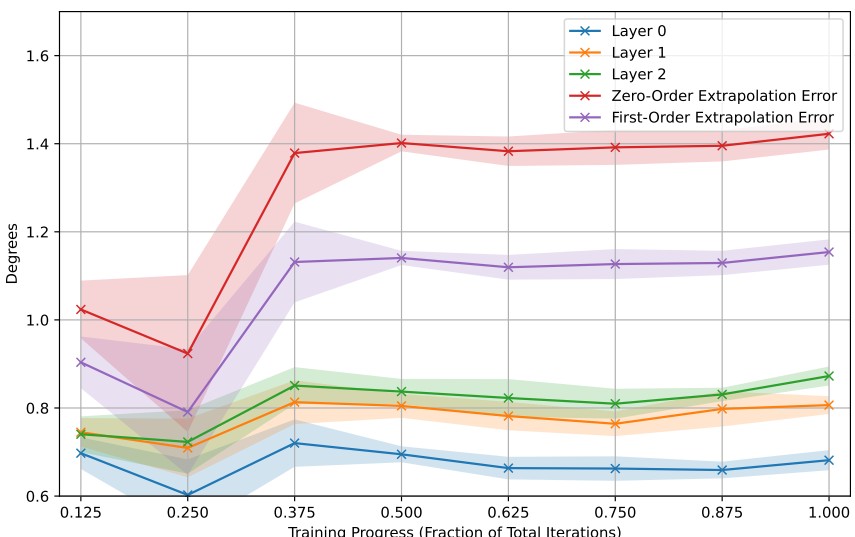

Figure 8: Errors of dynamics prediction based on representations from different layers, *Locomotion* task.

---

**Algorithm 2** Dynamics Knowledge Probing in Vanilla Policy Networks

**Require:** policy network models $\{\theta_i\}_{i=0}^M$, representation extraction functions $\{f_j\}_{j=1}^N$, required dataset size $N$, simulation environment $\mathcal{E}$, lightweight network $g$ parameterized by $\psi$
1: **for** each $i$ in $1, 2, ..., M$ **do**      ▷ each policy network model under our investigation
2:      $\theta \leftarrow \theta_i, \mathcal{D}_i \leftarrow \{\}, t \leftarrow 0$
3:      **repeat**
4:          $o_t \sim p_{\pi_\theta}(o_t)$
5:          Get the mean action $a_t \leftarrow \mu_{\theta_\mu}(o_t)$
6:          Step the environment $\Delta q_{t+1} \leftarrow \mathcal{E}.\text{step}(a_t)$
7:          $\mathcal{D}_i.\text{append}\big((o_t, \Delta q_{t+1})\big)$
8:          $t \leftarrow t + 1$
9:      **until** $\text{len}(\mathcal{D}_i) \geq N$
10:      **for** each $j$ in $1, 2, ..., N$ **do**      ▷ for latent representation extracted after each activation layer
11:          $f \leftarrow f_j$
12:          $\mathcal{D}_i^l \leftarrow \{(l_t, o_t) \mid l_t = f_{\theta_\mu}(o_t), o_t \in \mathcal{D}_i\}$
13:          Initialize lightweight MLP $g_\psi$
14:          Fit $g_\psi$ optimizing following target

$$\psi^\star = \arg\min_\psi \frac{1}{N} \sum_{l_t \in \mathcal{D}_i^l} \|g_\psi(l_t, o_t) - \Delta q_{t+1}\|_1$$

15:          evaluate final error

$$\epsilon_{ij} = \frac{1}{N} \sum_{l_t \in \mathcal{D}_i^l} \|g_\psi^\star(l_t, o_t) - \Delta q_{t+1}\|_1$$

16:      **end for**
17: **end for**
18: **return** $\{\epsilon_{ij}\}_{i=1,...,M}^{j=1,...,N}$

## A.3 EXPLORATION-BASED DATA COLLECTION IMPLEMENTATION DETAILS

The exploration is guided by both intrinsic reward and a set of extrinsic reward terms. We define the intrinsic reward as the standard deviation of all predictions generated by each individual PIDM model within the ensemble:

$$\hat{a_t}^j = I_{\xi_j}(x_{t-K:t+1}, a_{t-K:t-1}) \tag{1}$$

$$\bar{a}_t = \sum_{i=1}^{N_I} \hat{a_t}^j \tag{2}$$

$$\sigma_t = \sqrt{\sum_{i=1}^{N_I} (\hat{a_t}^j - \bar{a}_t)^2} \tag{3}$$

$$r_t^i = \min(c_{\text{ir}}\sigma_t, r_{\text{i\_max}}) \tag{4}$$

where $\{\xi_j\}_{j=1}^{N_I}$ denotes $N_I$ individual PIDM dynamics model in the ensemble, $\hat{a_t}^j$ denotes the action inference by the $j$-th PIDM dynamics model, $\sigma_t$ denotes the standard deviation of predictions, $r_t^i$ indicates the intrinsic reward at timestep $t$, $c_{\text{ir}}$ is the intrinsic reward scaling factor, and $r_{\text{i\_max}}$ is the intrinsic reward clipping threshold. The two hyperparameters are tuned empirically, indicated in Table 4. The set of extrinsic reward terms of ANYmal D are shown in Table 5. Extrinsic reward terms of other two embodiments can be found in the supplementary code.

The data collection pipeline is outlined in Algorithm 3, and the bootstrap training of models in the ensemble is described in Algorithm 4. Hyperparameters are listed in Table 4.

---

**Algorithm 3** PIDM-Ensemble Exploration-based Data Collection

---

**Require:** policy network parameters $\theta$, value network parameters $\phi$, minimum required dataset size $N_{\mathcal{D}}$, parallel environments $\mathcal{E}_{N_{\mathcal{E}}}$ with $N_{\mathcal{E}}$ sub-environments, PIDM architecture $I$, randomly initialized PIDM weights $\xi$, maximum iteration number $N$, PIDM ensemble retrain interval $k$

1: Initialize $\mathcal{E}$
2: $\mathcal{D} \leftarrow \{\}$
3: **for** each $i$ in $0, 1, 2, ..., N-1$ **do**                                                    ▷ PPO iterations
4:      $\{s_{t-K:t}, a_{t-K:t}, r_{t-K:t}^e, x_{t-K:t}\} \leftarrow \text{EnvironmentSteps}(\mathcal{E}, \pi_\theta)$
                      ▷ $x_{t-K:t}$ denotes noise-free observations, and $r_{t-K:t}^e$ denotes extrinsic rewards
5:      $\mathcal{D} = \mathcal{D} \cup \{x_{t-K:t}\}$
6:      **if** $\text{Size}(\mathcal{D}) \geq N_{\mathcal{D}}$ **then**
7:          **if** $i \mod k = 0$ **then**
8:              $\mathcal{D}_{\text{train}} \leftarrow \text{RandomSplit}(\mathcal{D})$ such that $\text{Size}(\mathcal{D}_{\text{train}}) = N_{\mathcal{D}}$
9:              $\{\xi_j\}_{j=1}^{N_I} \leftarrow \text{TrainEnsemble}(\{\xi_j\}_{j=1}^{N_I}, \mathcal{D}_{\text{train}})$
10:         **end if**
11:         $r_{t-K:t}^i \leftarrow \text{GetIntrinsicReward}(s_{t-K:t}, a_{t-K:t}, r_{t-K:t}^e, x_{t-K:t})$
12:         $\theta \leftarrow \text{PPOUpdateActor}(\theta, s_{t-K:t}, a_{t-K:t}, r_{t-K:t}^e, r_{t-K:t}^i)$
13:     **else**
14:             $r_{t-K:t}^i \leftarrow 0$
15:     **end if**
16:     $\phi \leftarrow \text{PPOUpdateCritic}(\phi, s_{t-K:t}, r_{t-K:t}^e, r_{t-K:t}^i)$
17: **end for**
18: **return** $\mathcal{D}$

---

---

**Algorithm 4** `TrainEnsemble` Function

---

**Input:** PIDMerse dynamics model weigths $\{\xi_j\}_{j=1}^{N_I}$, PIDMerse dynamics training set $\mathcal{D}_{\text{train}}$,

1: **for** each $i$ in $1, 2, ..., N_I$ **do**
2:     $\mathcal{D}_{\text{Train}}^j \leftarrow \{\}$
3:     $N \leftarrow \text{Size}(\mathcal{D}_{\text{Train}})$
4:     **for** each $n$ in $1, 2, ..., N$ **do**
5:         $\mathcal{D}_{\text{Train}}^j.\text{Append}\big(\text{SampleUniform}(\mathcal{D}_{\text{Train}})\big)$         ▷ Sample with replacement
6:     **end for**
7:     $\xi_j \leftarrow \text{Train}(\xi_j, \mathcal{D}_{\text{Train}}^j, I)$         ▷ Normal supervised training
8: **end for**
9: **return** $\{\xi_j\}_{j=1}^{N_I}$

---

| Item | Values |
|---|---|
| Ensemble size | 5 |
| Max iterations | 800 |
| Retrain Interval | 10 |
| Retrain epochs | 5 |
| Intrinsic reward scaling factor $c_{\text{ir}}$ | 10 |
| Intrinsic reward clipping threshold $r_{\text{i\_max}}$ | 30 |

Table 4: Exploration-based data collection hyperparameters.

| Term | Equation | Weight |
|---|---|---|
| feet air time | $\sum_{\text{foot}} \mathbf{1}_{\text{first\_contact}} \max\left(T, T_{\max}\right)^2$ | 400 |
| collision penalty | $\mathbf{1}_{\text{collision}}$ | $-5.0$ |
| joint torques | $|\tau|^2$ | -2e-5 |
| joint velocities | $|\dot{q}|^2$ | -5e-2 |
| joint acceleration | $|\ddot{q}|^2$ | -5e-6 |
| action magnitude | $|a_t|^2$ | -0.01 |
| action smoothing | $|a_{t-1} - a_t|^2$ | -0.01 |
| termination | $\mathbf{1}_{\text{termination}}$ | -80 |

Table 5: Extrinsic reward terms in exploration-based data collection for ANYmal-D.

## A.4 PIDM AND PRETRAINING IMPLEMENTATION DETAILS

PIDM is of a modular architecture consists of multiple MLPs serving different purposes. Every sub-module is implemented by an MLP. For *Action History Encoder*, *Proprioception History Encoder*, *Delta Encoder*, and *Intention Encoder*, they encode single-timestep input of various modalities into one embedding with a unified embedding dimension. The embeddings from various modalities and timesteps are then concatenated and fed into the PIDM backend, which synthesize the inputs and output an action embedding of embedding dimension. *Action Decoder* and *Action Synthesizer* decodes action signal from tensors of corresponding dimensions. The output of *Action Decoder* is processed through a Sigmoid activation layer and renormalized to the range $[-2.5\text{rad}, 2.5\text{rad}]$.

Pretrain is implemented via supervised training on the pretrain dataset. In the pretraining of ANYmal D embodiment, two techniques are applied to enhance robustness of the model:

- **Symmetry augmentation.** To effectively leverage the symmetry property of ANYmal robot we use, for each batch of training data in every epoch, we randomly divide the batch into 4 minibatches with equal number of samples. Then, they respectively go 1)

| Architecture | Item | Values |
|---|---|---|
| PIDM | Input history timesteps ($K$) | 4 |
| | Action history encoder | [128] |
| | Proprioception history encoder | [128] |
| | Delta encoder | [128] |
| | Action decoder | [128] |
| | Embedding dimension | 128 |
| | Backend | [512, 256, 128] |
| | Activation function | ELU |
| | Loss function | L1 loss |
| | Batch size | 1024 |
| | Optimizer | AdamW |
| | Learning rate | 1e-3 |
| | Training epochs | 260 |
| PIDM (RL-Blind) | Intention encoder | [128, 128, 128] |
| | Action synthesizer | [128, 128, 128] |
| PIDM (RL-Perceptive) | Intention encoder | [512, 256, 128] |
| | Action synthesizer | [512, 256, 128] |

Table 6: PIDM architecture hyperparameters. The values enclosed in square brackets indicate the number of layers and number of hidder units per layer in the corresponding MLP modules. *PIDM (RL-Blind)* hyperparameter set is used in blind tasks (locomotion, pedipulation), while *PIDM (RL-Perceptive)* hyperparameter set is used in perceptive tasks (parkour walk, crouch, jump, climb up, climb down).

> unchanged, 2) through x-axis symmetry transform, 3) through y-axis symmetry transform, or 4) sequentially go through both x-axis and y-axis transforms.

- **Noise addition.** To increase the model's resilience to perceptive noise, we add to every batch of training data in every epoch the noise vector sampled from the noise distribution identical to that in RL policy rollout. The scale of noise is identical across all tasks and can be found in e.g. Rudin et al. (2022). Notably, delta-state $\Delta x^*_{t+1}$ and ground truth action output label $a_t$ remain uncorrupted at all times.

Training a highly accurate PIDM poses great challenge. The reasons are:

- **Vast transition space.** With joint states, base twist, gravity, contact states, and terrain taken into consideration, the full state of a locomotion POMDP is notoriously large and thus hinders precise modeling. Recent work (Xu et al., 2025) has demonstrated success in learning a high-precision neural simulator using a lightweight GPT-2 model, which is orders of magnitude larger than our PIDM.

- **Partial observability.** The PIDM only has access to noisy proprioceptive observation, without any privileged information about randomized physical properties that are used to facilitate sim-to-real transfer.

- **Lack of inductive bias in MLP** architectures might not make them the best-suited option for analysing time-series data, despite their prevalent use in RL (Bachmann et al., 2023).

### A.5 PIDM Implementation Details

PIDM hyperparameters are also shown in Table 6.

### A.6 Quantitive Experiments Details

Original works are referred to as "vanilla MLP", and references are listed in Section 5.1. Most POMDP configurations and RL hyperparameters are kept the same with the original works to the maximum possible, and details can be found in the corresponding references.

| Group | Members | Tasks | | Models | |
|---|---|---|---|---|---|
| | | Blind | Perceptive | Vanilla MLP | PIDM |
| Proprioception | base linear velocity base angular velocity projected gravity vector joint position joint velocity | ✓ | ✓ | single timestep | multiple timesteps |
| Exteroception | height scan | ✗ | ✓ | - | - |
| Last action | joint action | ✓ | ✓ | single timestep | multiple timesteps |
| Command | task specific command | ✓ | ✓ | ✓ | ✓ |

Table 7: **Observation space configuration.** "Blind" tasks: locomotion, pedipulation. "Perceptive" tasks: parkour walk, climb up, climb down, crouch, and jump.

### A.6.1 SUMMARY OF OBSERVATION SPACE, ACTION SPACE AND SIMULATION

**Observation space.** All possible components of observation space are listed in Table 7. To summarize, there are 2 types of variation in the composition of observation space across all the runs: a) for the blind tasks (locomotion, pedipulation), policies do not have access to exteroception (height scans), while exteroception is included in the rest perceptive tasks. b) Vanilla MLPs do not have access to history proprioceptions as in the original works, while the inputs to PIDM based models contain history proprioceptions. The necessity of including a history of proprioception for PIDM is mainly due to the absense of terrain information and contact state in proprioception, and the presence of noise and domain randomization techniques in the training process (during both exploration-based data collection and training of task-specific policies). Therefore, it would be inappropriate to fit the PIDM with only one single frame of current proprioceptive state, due to the fact that one certain proprioception can be observed in a range of actual full states in the POMDP. In addition, there is a line of works (Ji et al., 2022; Portela et al., 2025) that suggest that proprioception history is informative and a number of useful values (e.g. foot height, contact probability, end effector force) can be estimated from it.

**Action space.** The action space is the target joint position (relative to default joint positions) command that will be sent to actuator-nets (Hwangbo et al., 2019) of ANYDrive 4.0.

**Simulation.** All training is performed in Isaac Lab (Mittal et al., 2025) using 4,096 parallel environments, each running 24 simulation steps per RL iteration, for a total of 98,304 environment steps per iteration. Each environmental step corresponds to 5 ms in real time and is computed using 4 physics solver steps.

### A.6.2 MODIFICATIONS

Following modifications are made to ensure easier and fair comparison.

**Unified collision model.** The collision model of ANYmal-D of all tasks is unified as that in the main branch of Isaac Lab 2.2.

**Pedipulation.** We change the action space of Pedipulation task from relative joint action space to absolute joint action space, to make it consistent with all other tasks. The curriculum (gradual expansion of the space where command is sampled) is removed because it makes comparing rewards from different stages/runs not appropriate. After removal of curriculum, the environment configuration is static and identical to that of the maximum difficulty in curriculum in original work.

**Parkour tasks.**

| Experiment | Task | Method | Number of Failed in 5 Runs |
|---|---|---|---|
| Quantitative Experiments | Walk
Crouch
Jump | PIDM (Random Init)
PIDM (Random Init)
PIDM (Random Init) | 1
1
3 |
| Ablation (Actor-Critic) | Climb Down | PIDM (Pretrained Actor only) | 3 |
| Ablation (Data Source) | Climb Up | PIDM (Pedipulation Data) | 2 |

Table 8: Failed runs. For those mentioned configurations, performance metrics are aggregated with the remaining successful runs. Notably there is no entry of the mainly proposed method.

- Fixed-step curriculum events are removed from *Climb Up* and *Jump*, for that the curriculum steps in these 2 tasks are extremely sensitive to the timing of trigger, which interferes with the learning once the architecture of networks changes.

- Adaptive terrain difficulty curriculum is carried over from the original work, but instead of initializing all parallel environments randomly at one difficulty level, they are all initialized at the lowest difficulty level in our experiments. This is for cleaner plot and is checked to have no visible impact on training dynamics, since almost all environments initialized with random difficulty level fall back to the lowest difficulty after a couple of iterations.

- Symmetrical augmentation (Mittal et al., 2024) is carried over from the original work. One critical observation is, adding symmetry loss does not significantly alter the training dynamics of vanilla MLP, but considerably improve the stability of RL training of PIDM, especially the randomly initialized variant. So in every experiments of every method, a symmetry loss weighted by 0.2 is added. On top of this, due to the failure-prone nature of learning highly dynamic skills, some configurations are still vulnerable to producing failed runs. All configurations that did not succeed every of 5 independent runs are listed in Table 8.

### A.6.3 PERFORMANCE INDICATOR

The choice of main performance indicator varies across tasks. Since the blind tasks (pedipulation, locomotion) are completely curriculum-free, we directly use the mean reward as the performance metric. However, since the adaptive (progress-based) terrain difficulty curriculum exist in all parkour tasks, the mean reward curves can not be directly taken as performance metric of policies because the evolving terrain difficulty. Therefore, we use the *Curriculum Progress*, indicated by the average of maximum terrain difficulty reached over all sub-environments as the way to assess learning performance.

### A.6.4 MORE REINFORCEMENT LEARNING TRAINING CURVES CORRESPONDING TO PRESENTED QUANTITATIVE RESULTS

See Figure 9.

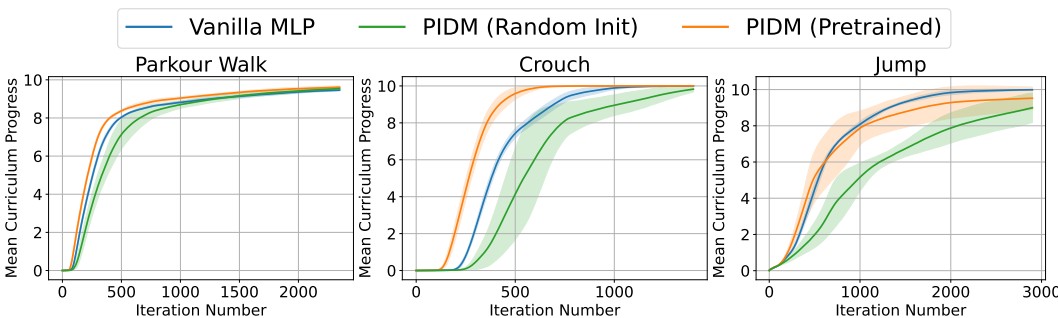

Figure 9: Evolution of the main performance metric during training for Parkour Walk, Pedipulation, Crouch and Jump tasks with ANYmal D. The shaded areas denote standard deviations across five seeds.

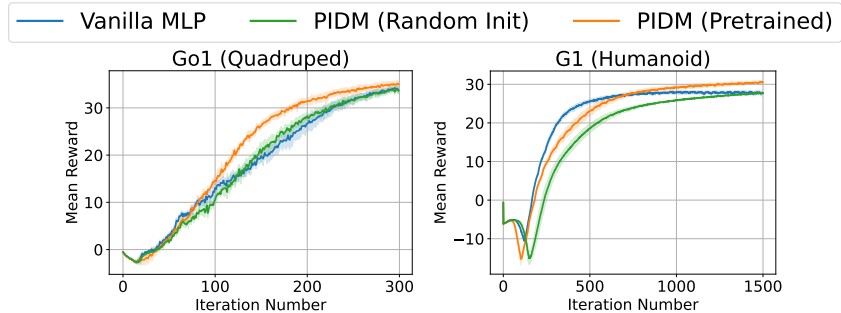

Figure 10: Evolution of the mean reward during learning velocity-tracking locomotion with Unitree Go1 (quadruped) and Unitree G1 (humanoid). The shaded areas denote standard deviations across five seeds.

## A.7 MORE ABLATION EXPERIMENTS TRAINING CURVES

The evolution of the performance indicator corresponding to results of Table 2 is shown in Figure 11 and curves of Table 3 is presented in Figure 12. Individual unstable configurations are listed in Table 8 as well.

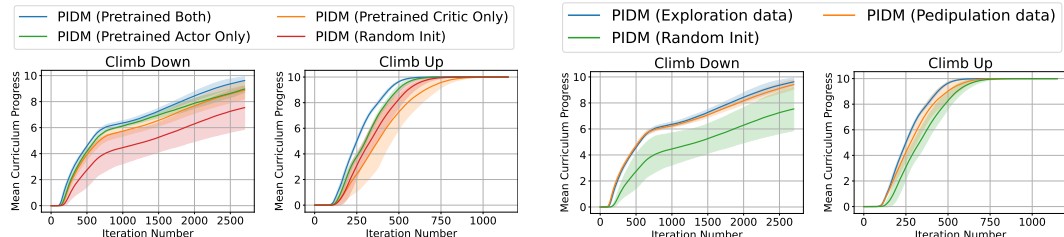

Figure 11: Ablation results for pretraining the actor vs. the critic components.

Figure 12: Ablation results comparing different sources of pretraining data.

## A.8 STUDY OF RELATION BETWEEN ERROR LEVEL OF PIDM AND RL PERFORMANCE

To verify the positive correlation between the accuracy of pretrained PIDM and resulted gain in RL performance, we study 3 PIDM model checkpoints from pretraining which respectively produces normalized error level of 75%, 60%, and 40%, as shown in Figure 13. The checkpoint that produces normalized error of 40% is the one that used to present the main quantitative results.

We benchmarked the 3 model checkpoints, along with the randomly initialized PIDM in the Climb Down and Climb Up tasks and plotted the performance indicator curves in Figure 14. The results

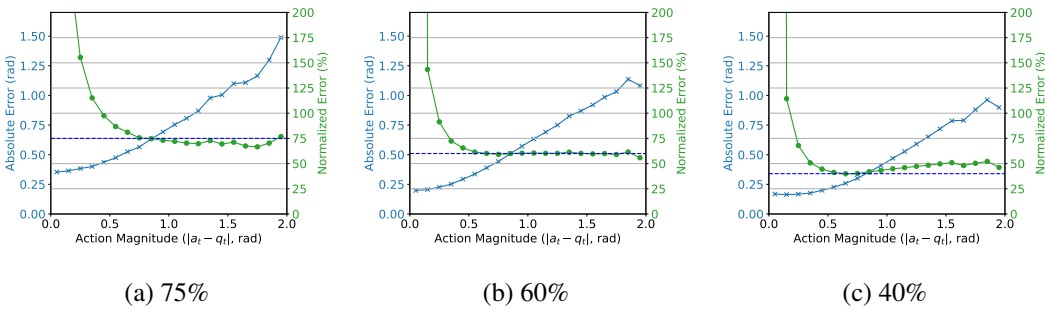

(a) 75%    (b) 60%    (c) 40%

Figure 13: PIDM error levels of the 3 checkpoints used to study the relation between PIDM error and RL performance, indicated by absolute and normalized joint errors.

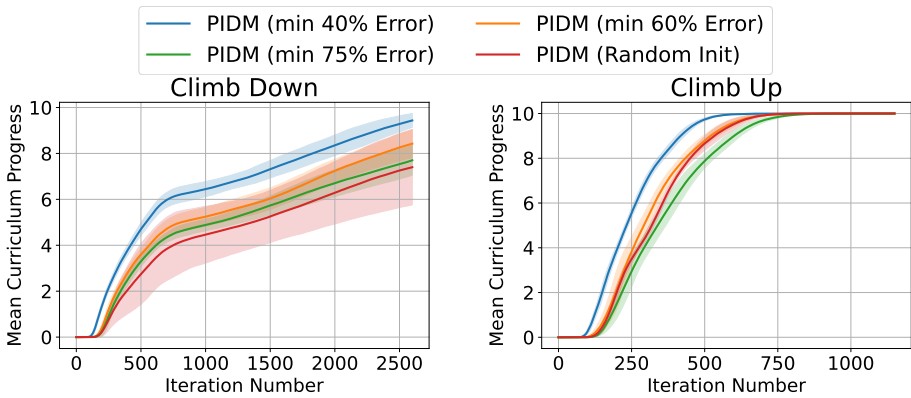

Figure 14: Study of relation between error Level of PIDM and RL performance.

suggest that in both tasks PIDM of 40% performs the best among all variant, followed by the model of 60% error. PIDM of 75% and randomly initialized PIDM are relatively underperforming the other two. These results suggest that the more accurate PIDM tends to yield better performance in downstream RL tasks. This finding might also hint that larger benefits are possible if higher accuracy can be attained using larger model and more modern architectures to implement PIDM. We leave this for future work.

### A.9    WEIGHT UPDATE MAGNITUDE OBSERVATION

The average update magnitude of each individual parameter is firstly averaged within each linear layer, producing a update magnitude metric for the layer. Then, the update magnitude of a submodule is given by the mean update magnitude metric of all layers it contains. Plots are shown in Figure 15.

### A.10    SIM-TO-REAL TRANSFER DEMONSTRATION

We have deployed the policy *Locomotion* on a real ANYmal D robot. The video is attached in the supplementary material.

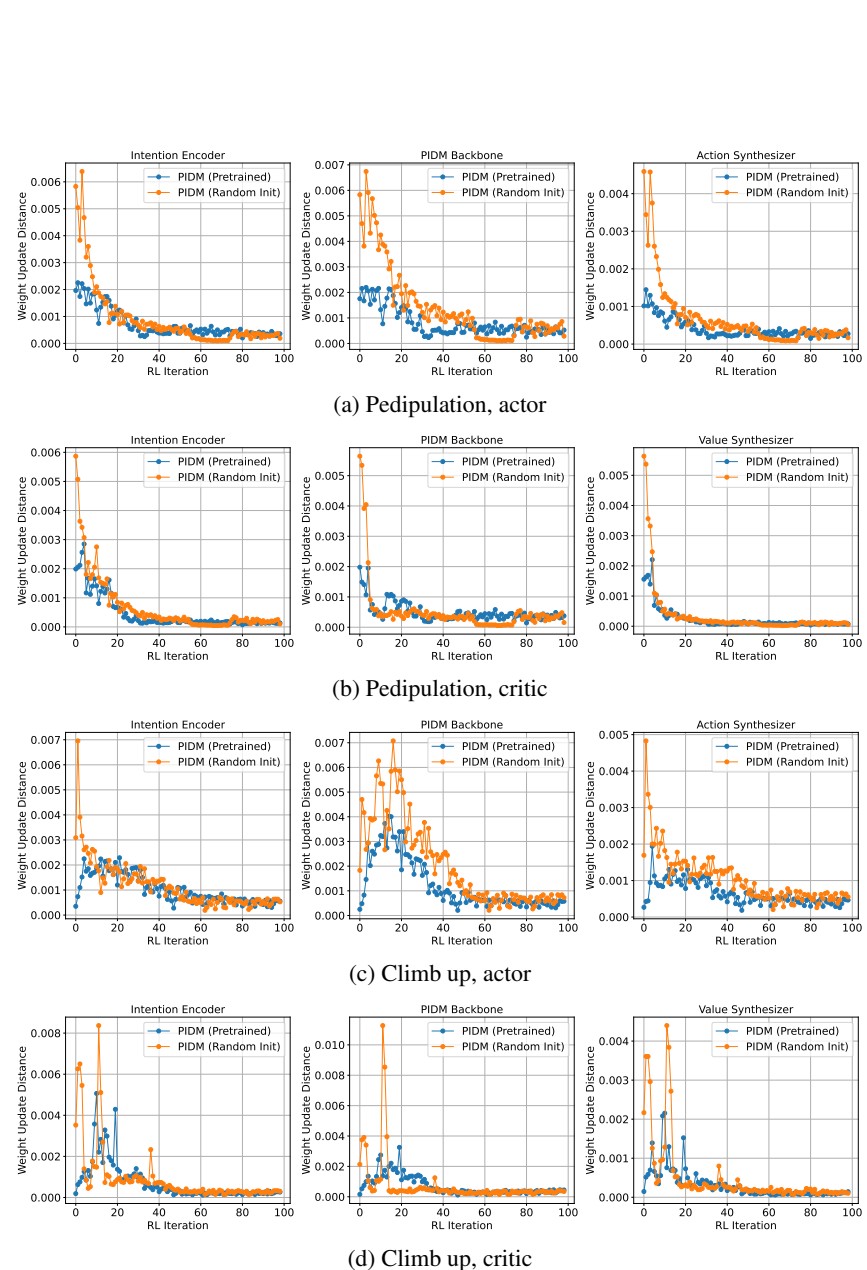

(a) Pedipulation, actor

(b) Pedipulation, critic

(c) Climb up, actor

(d) Climb up, critic

Figure 15: Network weight update magnitude comparison.

