# OpenReview forum: "Pretraining in Actor-Critic Reinforcement Learning for Robot Motion Control"
_ICLR.cc/2026/Conference — Submitted to ICLR 2026_

### Official Review · Reviewer_J3PS · 2025-10-29

**Soundness:** 2
**Presentation:** 3
**Contribution:** 3
**Rating:** 4
**Confidence:** 4

**Summary:**

The paper presents a way to pretrain RL policies by introducing an inverse dynamics model in the actor-critic architecture. The pretraining is valid for a single embodiment with arbitrary tasks.
The algorithm is evaluated in simulation on quadruped locomotion tasks.

**Strengths:**

- The problem setting is interesting and timely.
- The proposed method is well motivated and generally applicable.
- The evaluations are chosen well and clearly show the benefit of the proposed method.

**Weaknesses:**

- The evaluation is limited to a single embodiment.
The idea of an inverse dynamics model is applicable to many (if not all) embodiments found in RL benchmark problems. Additional results on different embodiments, for example those found in Gymnasium, would strengthen the paper significantly. With the presented results it is difficult to judge whether and for which systems the proposed pretraining is helpful.

- The method is the reliance on well curated prior data set.

I would hypothesis that the quality of the inverse dynamics model and therefore the pretraining success highly depends on the training data. In fact, the collection of the dataset seems to be a major effort that includes many hyperparameters (including uncertainty quantification, domain randomization, and reward tuning). There is only a single ablation w.r.t. the quality of the data set.

- Missing baselines. Model-based and offline RL can both be used for pre-training (for example [1]. There is generally a big body of work on "Offline-to-online reinforcement learning" [2] that can serve as baseline.

[1] Nakamoto, M., Zhai, S., Singh, A., Sobol Mark, M., Ma, Y., Finn, C., ... & Levine, S. (2023). Cal-ql: Calibrated offline rl pre-training for efficient online fine-tuning. Advances in Neural Information Processing Systems, 36, 62244-62269.

[2] Xie, Z., Lin, Z., Li, J., Li, S., & Ye, D. (2022). Pretraining in deep reinforcement learning: A survey. _arXiv preprint arXiv:2211.03959_.

**Questions:**

-  For Figure 3 and Section 4.5: I assume the figure and section only describes the actor and the critic gets a 'value synthesizer' head instead of an 'action synthesizer'? Are there other difference between the value and critic? Do they share the same backbone?

---

> ### Author Response · Authors · 2025-11-21
>
> > **Weakness 1** The evaluation is limited to a single embodiment. The idea of an inverse dynamics model is applicable to many (if not all) embodiments found in RL benchmark problems. Additional results on different embodiments, for example those found in Gymnasium, would strengthen the paper significantly. With the presented results it is difficult to judge whether and for which systems the proposed pretraining is helpful.
> >
>
> Thank you for your insightful feedback. We are currently working on showcasing the performance of our method with more embodiments, and hopefully we would be able to present it by the end of discussion period.
>
> > **Weakness 2** The method is the reliance on well curated prior data set.
> I would hypothesis that the quality of the inverse dynamics model and therefore the pretraining success highly depends on the training data. In fact, the collection of the dataset seems to be a major effort that includes many hyperparameters (including uncertainty quantification, domain randomization, and reward tuning). There is only a single ablation w.r.t. the quality of the data set.
> >
>
> From our interpretation on the dataset ablation (Table 3), we would draw the opposite conclusion that the method is not picky on dataset, as the initial stages of RL training of pedipulation can also yield notable performance gains. About the design of dataset collection MDP: the noise levels and stability reward terms are largely taken from the original works of task-specific policy training and have not been substantially altered; the main part required to tune is the formulation and weight of the intrinsic exploration reward, which we deem to be of reasonable workload but not heavy.
>
> > **Weakness 3** Missing baselines. Model-based and offline RL can both be used for pre-training (for example [1]. There is generally a big body of work on "Offline-to-online reinforcement learning" [2] that can serve as baseline.
> >
>
> We would like to clarify that our work addresses a different problem compared to the majority of model-based and offline-to-online RL methods. We aim to provide task-agnostic weights initialization for all possible downstream tasks of the specific embodiment. The unknownness and possible variation of downstream MDPs determine that it is impossible to include task-specific reward signal in the pretraining dataset, which does not fulfill the preliminaries of [1]. We have reviewed related literatures extensively but have not found works that fullfill the same requirements as ours:
>
> 1. The method does not need reward signal of the task-specific downstream MDPs to be present in the pretraining dataset.
> 2. The pretraining shows successful transfer in multiple downstream tasks.
> 3. Except the addition of pretraining phase, the method does not alter the widely adopted PPO locomotion learning pipeline.
>
> > **Question** For Figure 3 and Section 4.5: I assume the figure and section only describes the actor and the critic gets a 'value synthesizer' head instead of an 'action synthesizer'? Are there other difference between the value and critic? Do they share the same backbone?
> >
>
> Totally correct, the critic gets a 'value synthesizer' head that outputs scalar estimated value, which is optimized by MSE loss, and there is no other difference from the actor. The actor and the critic do not share the same backbone.

---

### Official Review · Reviewer_HATj · 2025-10-30

**Soundness:** 3
**Presentation:** 3
**Contribution:** 3
**Rating:** 4
**Confidence:** 3

**Summary:**

The paper proposes a pretraining method for actor–critic RL in robot motion control. A Proprioceptive Inverse Dynamics Model (PIDM) is trained on task-agnostic exploration data and used to initialize both actor and critic networks in PPO. On seven quadruped tasks, this warm-start improves sample efficiency by 40.1% and final performance by 7.5% over random initialization.

**Strengths:**

* Clear idea and solid execution.
* Experiments are broad and well-controlled, with ablations and weight-update analyses that convincingly show consistent gains.

**Weaknesses:**

* Limited to one embodiment (ANYmal-D); generalization to other robots remains untested.
* The PIDM’s low predictive accuracy (40–50%) raises questions about what is actually transferred.
* Missing stronger model-based baselines (like TD-MPC2).

**Questions:**

* Would results hold for manipulators or bipeds?
* Is the benefit due to dynamics knowledge or regularization?
* Could recurrent or attention-based PIDMs improve transfer?
* Is there any multi-modality issue when fitting one model with multiple source data?

---

> ### Author Response · Authors · 2025-11-21
>
> > **Weakness 1** Limited to one embodiment (ANYmal-D); generalization to other robots remains untested.
> **Questions 1** Would results hold for manipulators or bipeds?
> >
>
> Thank you for the feedback, and we are currently working on showcasing the performance of our method with more embodiments. Hopefully we’ll present it by the end of discussion phase.
>
> > **Weakness 2** The PIDM’s low predictive accuracy (40–50%) raises questions about what is actually transferred.
> **Question 2** Is the benefit due to dynamics knowledge or regularization?
> >
>
> We understand that the prediction accuracy constitutes a reasonable concern, and we analysed the reasons in Appendix A.4. In short, a recent successful work [1] in training highly accurate forward dynamics model requires model size larger by several magnitudes, but traditionally actor and critic networks in motion policy training are extremely lightweight compared to other research fields of machine learning, which means that we can not substantially increase the model size if we would like to adopt the methods from prior works as-is (also such large models are known to make RL hard [2, 3]). Additionally, our model need to be robust so it is under considerable influence of noise and randomization, which also contributes to a higher level of error. However, we believe that a predictive accuracy of 40%-50% already represents considerable learned patterns compared with pure random initialization, which provides a good start point for RL process. Regularization that resulted from pretraining can be also contributing to the performance gain in later RL process, though it’s hard to quantify. Furthermore, we would work on a study that shows how prediction accuracy correlates with performance gain to better explain if time permits.
>
> > **Weakness 3** Missing stronger model-based baselines (like TD-MPC2).
> >
>
> We would like to clarify that our work addresses a different problem compared to the majority of model-based RL methods, including TD-MPC2. We aim to provide a task-agnostic weights initialization for all possible downstream tasks of the specific embodiment. The unknownness and possible variation of downstream MDPs determine that it is impossible to include task-specific reward signal in the pretraining dataset. TD-MPC2 clearly requires the task MDP is fully known and free to explore in advance, which is not available under the setup of our problem.
>
> > **Question 3** Could recurrent or attention-based PIDMs improve transfer?
> >
>
> We suppose that PIDMs with higher accuracy would provide larger performance gains, so if trained right, architectures with larger capacity like recurrent or attention-based PIDMs can potentially bring even better performance. Nonetheless, to quantify this gain, RL processes need to be first tuned to randomly initialized such actors and critics of large size to provide meaningful baselines, which are known to be hard [2, 3], have not existed in prior works and would constitute heavy workloads. We leave these directions for future works.
>
> > **Question 4** Is there any multi-modality issue when fitting one model with multiple source data?
> >
>
> We are not sure if we have understood this question. If “multi-modality issue” refers to the multiple modalities incurred by the variation of terrains, noise, and domain randomization levels, that multi-modality phenomenon does exist, and should be contributing to the difficulty of achieving high accuracy with the trained PIDM. Nonetheless the results show that the presence of such multi-modality does not create issues that hinder improvements in RL processes.
>
> [1] Xu, Jie, Eric Heiden, Iretiayo Akinola, Dieter Fox, Miles Macklin, and Yashraj Narang. "Neural robot dynamics." *arXiv preprint arXiv:2508.15755* (2025).
>
> [2] Ota, Kei, Devesh K. Jha, and Asako Kanezaki. "Training larger networks for deep reinforcement learning." *arXiv preprint arXiv:2102.07920* (2021).
>
> [3] Li, Wenzhe, Hao Luo, Zichuan Lin, Chongjie Zhang, Zongqing Lu, and Deheng Ye. "A survey on transformers in reinforcement learning." *arXiv preprint arXiv:2301.03044* (2023).

---

### Official Review · Reviewer_RrdJ · 2025-11-01

**Soundness:** 2
**Presentation:** 2
**Contribution:** 2
**Rating:** 2
**Confidence:** 3

**Summary:**

The paper proposes a pretraining–finetuning scheme for actor–critic RL in robot motion control. The authors first collect task‑agnostic exploration data and train a Proprioceptive Inverse Dynamics Model (PIDM). The pretrained PIDM weights are then used to initialize both actor and critic via a modular architecture, aiming to warm‑start PPO without changing task formulations or hyperparameters. On seven ANYmal‑D tasks (locomotion, pedipulation, and five parkour skills), the method reports an average +7.5% final performance increase and 40.1% fewer iterations.

**Strengths:**

Well written and easy to follow; the modular design makes integration into PPO straightforward.

Consistent gains across seven tasks with clear metrics

Ablations (actor vs. critic vs. both; exploration vs. task data) and probing analyses provide insight into why the initialization helps

Code shared with intent to open‑source; good reproducibility posture

**Weaknesses:**

Comparisons are limited: beyond a vanilla MLP and “PIDM random init” there are no direct baselines to other pretraining/weight‑sharing approaches in RL. This makes it hard to position the gain relative to prior art.

Missing control for “pretrain the policy itself”: it would be important to compare against pretraining a standard actor–critic on the same exploration data (e.g., behavior‑cloning warm start or supervised inverse‑model head inside the policy) to isolate the benefit of the PIDM module versus generic weight initialization.

Practical clarity: the repository would benefit from clearer “contribution scripts”/entry points for (i) data collection, (ii) PIDM pretraining, and (iii) task fine‑tuning to ease reproduction.

**Questions:**

How does performance change if you directly pretrain the actor–critic on the same exploration dataset?

Can you provide adapted baselines from prior pretraining/weight-sharing work in RL to strengthen positioning? Even if not perfectly matched, a careful adaptation in Isaac Lab would be informative.

PIDM’s normalized error is ~40–50% overall. Do higher‑accuracy PIDMs (e.g., via longer training or temporal models) correlate with larger RL gains? A small study varying pretraining budget could clarify whether accuracy or structure (i.e., the inductive decomposition) drives the benefit.

What happens if you freeze the PIDM backbone initially (or for a set number of iterations) and only train the Intention encoder/Action synthesizer? This could separate “good initialization” from “continual adaptation.”

---

> ### Author Response · Authors · 2025-11-21
>
> > **Weakness 1** Comparisons are limited: beyond a vanilla MLP and “PIDM random init” there are no direct baselines to other pretraining/weight‑sharing approaches in RL. This makes it hard to position the gain relative to prior art.
> **Question 2** Can you provide adapted baselines from prior pretraining/weight-sharing work in RL to strengthen positioning? Even if not perfectly matched, a careful adaptation in Isaac Lab would be informative.
> >
>
> We would like to emphasize that “no direct baselines to other pretraining/weight‑sharing approaches in RL” is due to the lack of existing works that can handle our targeted problem. We are not aware of prior art that tackles our problem setting, and would appreciate if you could give references to the prior art to your knowledge.
>
> > **Weakness 2** Missing control for “pretrain the policy itself”: it would be important to compare against pretraining a standard actor–critic on the same exploration data (e.g., behavior‑cloning warm start or supervised inverse‑model head inside the policy) to isolate the benefit of the PIDM module versus generic weight initialization.
> **Question 1** How does performance change if you directly pretrain the actor–critic on the same exploration dataset?
> >
>
> We would like to clarify that we aim to provide a task-agnostic weights initialization for all possible downstream tasks of the specific embodiment. The unknownness and possible variation of downstream MDPs determine that it is impossible to include task-specific reward signal in the pretraining dataset, which makes “pretraining a standard actor–critic on the same exploration data”, “behavior‑cloning warm start” infeasible. If “generic weight initialization” is referring to the widely used default random initialization, we believe that the benefit has been well showcased in Table 1; otherwise we would appreciate if you further explain what specific methods you referred to.
>
> > **Weakness 3** Practical clarity: the repository would benefit from clearer “contribution scripts”/entry points for (i) data collection, (ii) PIDM pretraining, and (iii) task fine‑tuning to ease reproduction.
> >
>
> Thank you for the feedback. We are working on improving our readme file to include pointers to the mentioned parts.
>
> > **Question 3** PIDM’s normalized error is ~40–50% overall. Do higher‑accuracy PIDMs (e.g., via longer training or temporal models) correlate with larger RL gains? A small study varying pretraining budget could clarify whether accuracy or structure (i.e., the inductive decomposition) drives the benefit.
> >
>
> If time permits we would work on a study that shows how different PIDM’s normalized error levels in percentage affect RL gains. Utilizing temporal models is mentioned as future work. While this work is a proof that this methodology does bring certain benefits, the ceiling of such method can be much higher and can potentially be attained with better model architectures. For the concern that it might be structure (i.e., the inductive decomposition) that drives the benefit, we are confident that the results in Table 1 excludes this possibility: as compared to the vanilla MLP,  randomly initialized PIDM is worse by 2.8% in final performance and 16.3% in sample efficiency (normalized by the performances of randomly initialized PIDM).
>
> > **Question 4** What happens if you freeze the PIDM backbone initially (or for a set number of iterations) and only train the Intention encoder/Action synthesizer? This could separate “good initialization” from “continual adaptation.”
> >
>
> We are not sure whether we have understood your proposed purpose of this experiment “separate ‘good initialization’ from ‘continual adaptation’”. In our opinion even with good initialization, continual adaptation is still absolute essential to learn task-specific policies, especially under the context that task-specific configuration (observation configuration, meaning of command etc.) and reward structure/signal do not exist in pretraining dataset. In our understanding, the need for continual adaptation does not contradict with good initialization.
>
> However, we did do experiments where we vary the number of initial iterations when the PIDM backbone is kept frozen, and the results show that the performance is negatively correlated to the number of iterations of freezing PIDM backbone (this experiment is not included in the manuscript). Therefore we eventually did not apply any freezing in the final proposed method.

---

### Official Review · Reviewer_QTfi · 2025-11-01

**Soundness:** 2
**Presentation:** 2
**Contribution:** 2
**Rating:** 4
**Confidence:** 3

**Summary:**

This paper introduces a method for utilizing pre-training in the context of reinforcement learning.
Specifically, it proposes to learn a task-agnostic Proprioceptive Inverse Dynamics Model (PIDM) from data collected by an exploration policy before the agent is trained on the actual task.
The PIDM is then used as part of the actor and critic networks during task-specific training.
They show that this approach improves both sample efficiency and final performance on 7 simulated locomotion tasks involving the ANYmal robot when trained with PPO.

**Strengths:**

- This work addresses an important challenge in reinforcement learning: improving sample efficiency through pre-training.
I believe pre-training could be an important step towards making RL more practical for real-world applications, especially in robotics.
By fully separating the pre-training phase from the task-specific training, the proposed method has the potential to reduce training time for a wide variety of tasks.

- Furthermore, the proposed method seems compatible with many existing RL algorithms, as it only proposes a modification to the actor and critic models and not to the training algorithm itself, except for the pre-training phase.

**Weaknesses:**

- In my opinion, the main weakness of this work is that the method is evaluated on a single type of robot (ANYmal) in simulation and for locomotion only. Yet, it is framed as a general method for pre-training in reinforcement learning for robot motion control. I believe that this kind of framing is not sufficiently supported by the experiments presented in the paper. Specifically, I would suggest the following improvements:
  - In the context of locomotion, the paper could benefit from evaluations on a wider variety of robots (e.g., see LocoMujoco [1] which provides a simulation of many different robots)
  - It should either be made clear that this method is intended for locomotion tasks, or the method should be evaluated on other types of tasks. In particular in manipulation tasks, the role of proprioceptive information may be less important than in locomotion tasks, and it would be interesting to see how the method performs in such a context.
  - In the conclusion, the authors state that "in contrast to related work, [their] modeling of the problem addresses the sim-to-real gap in robotics". However, no sim-to-real experiments are presented in the paper. I believe that either such experiments should be added, or this claim should be removed.
- Hypothesis 1, which states that neural network policies encode intended target states in the earlier layers and actions in the later layers is not convincingly supported by the experiments in Section 5.2 for two reasons:
  1. Layer 0 eventually surpasses layer 1 in prediction error, contradicting the statement that "the correlation between the future state diminishes as we progress deeper into the network".
  2. The observed effect could also have explanations different from Hypothesis 1. E.g., if the output actions are motor torques, then in order to reconstruct the delta angles from them, the current velocities are needed. Information about those might be present in the first layers of the network and not in the later layers. Hence, in my opinion, further analysis is needed to support Hypothesis 1.

[1] Al-Hafez, Firas, et al. "Locomujoco: A comprehensive imitation learning benchmark for locomotion." arXiv preprint arXiv:2311.02496 (2023).

**Questions:**

- What is the state and action space of the experiments with ANYmal?
- Why does the PIDM model require a history of proprioceptive states? Are joint velocities not included in x?
- In Figure 6, how many environment steps does each iteration amount to?
- How is the PIDM used in the critic?

---

> ### Author Response · Authors · 2025-11-21
>
> > **Weakness 1**
> >
> > - In my opinion, the main weakness of this work is that the method is evaluated on a single type of robot (ANYmal) in simulation and for locomotion only. Yet, it is framed as a general method for pre-training in reinforcement learning for robot motion control. I believe that this kind of framing is not sufficiently supported by the experiments presented in the paper. Specifically, I would suggest the following improvements:
> >     - In the context of locomotion, the paper could benefit from evaluations on a wider variety of robots (e.g., see LocoMujoco [1] which provides a simulation of many different robots)
> >     - It should either be made clear that this method is intended for locomotion tasks, or the method should be evaluated on other types of tasks. In particular in manipulation tasks, the role of proprioceptive information may be less important than in locomotion tasks, and it would be interesting to see how the method performs in such a context.
> >     - In the conclusion, the authors state that "in contrast to related work, [their] modeling of the problem addresses the sim-to-real gap in robotics". However, no sim-to-real experiments are presented in the paper. I believe that either such experiments should be added, or this claim should be removed.
>
> Thank you for your insightful feedback. We are currently working on showcasing the performance of our method in more scenarios, and on deploying the the policy for one task (“locomotion”) on real hardware. Hopefully we would be able to present it by the end of discussion period.
>
> > **Weakness 2**
> >
> > - Hypothesis 1, which states that neural network policies encode intended target states in the earlier layers and actions in the later layers is not convincingly supported by the experiments in Section 5.2 for two reasons:
> >     1. Layer 0 eventually surpasses layer 1 in prediction error, contradicting the statement that "the correlation between the future state diminishes as we progress deeper into the network".
> >     2. The observed effect could also have explanations different from Hypothesis 1. E.g., if the output actions are motor torques, then in order to reconstruct the delta angles from them, the current velocities are needed. Information about those might be present in the first layers of the network and not in the later layers. Hence, in my opinion, further analysis is needed to support Hypothesis 1.
> 1. We recognize that at iteration 4000 the layer 0 achieved slightly less error than layer 1. However we are aware that certain uncertainty exists within this experiment, e.g. the fitting process of lightweight MLP, which might cause the resulted value to fluctuate a bit around the true expectation. Though that specific point diverges from the expected trend, we believe this deviation does not considerably impact the overall pattern observed.
> 2. We agree this might also be a working explanation (by the way the output actions are target joint positions in our case). In fact, due to the compactness of a vanilla 4-layer MLP, it can be hard to precisely identify what information is contained in the intermediate representations. For our motivation, we are interested in how we can interpret an end-to-end learned policy in a modular way, and then identify reusable parts that we can pretrain prior to starting RL. From this point of view, the hypothesis inspired us to try to implement a modular architecture that has this analogy to vanilla MLPs, and the results have proved its efficacy. That is the reason why we want to share this experiment and our interpretation of it, rather than to perfectly prove Hypothesis 1.

---

> > ### Comment · Reviewer_QTfi · 2025-11-24
> >
> > Regarding weakness 2:
> >
> > 1. With how many seeds did you run this evaluation and on how many tasks? Could you add the standard deviation to this plot?
> >
> > 2. I think the fact that you are not actually proving Hypothesis 1 should be made clear in that case, and also, alternative explanations should be discussed. In particular because even at iteration 0, when the network is still random, the correlation between reconstruction error and layer number seems to be present. This means that it is just generally easier to reconstruct the target state from earlier hidden states, even for random neural networks.
> >
> > Furthermore, if the action space is target joint positions and the input space of the PIDM includes delta joint positions, have you looked into how much the PIDM deviates from just adding the deltas to the current joint angles to produce target joint positions? Also, how does your method compare to a vanilla MLP that produces delta joint positions instead of absolute joint positions? I mean it would be a very compelling argument for using PIDM if you can show that it, whenever necessary, intelligently produces target joint positions different from the intended target joint position by the policy in order to make other components of the state match.

---

> > > ### Author Response · Authors · 2025-12-03
> > >
> > > > Regarding weakness 2:
> > > >
> > > > 1. With how many seeds did you run this evaluation and on how many tasks? Could you add the standard deviation to this plot?
> > >
> > > We have improved the setup of the dynamics knowledge probing experiment to address your concerns:
> > >
> > > - To exclude the possibility that the worse prediction based on representations from deeper layers is resulted from insufficient information about current state, we now append the raw observation into the input to the lightweight prediction MLP, to always provide the same amount of information about the current state. Details are described in Appendix A.2.
> > > - We scale the experiment to 2 tasks, and 5 runs with different random seeds are performed in each task. Standard deviations, along with zero and first-order extrapolation errors are also provided for reference in Figure 4b and Figure 9.
> > >
> > > The updated experiment continues to show the same pattern of error levels of representations.
> > >
> > > > 2. I think the fact that you are not actually proving Hypothesis 1 should be made clear in that case, and also, alternative explanations should be discussed. In particular because even at iteration 0, when the network is still random, the correlation between reconstruction error and layer number seems to be present. This means that it is just generally easier to reconstruct the target state from earlier hidden states, even for random neural networks.
> > > >
> > >
> > > We have taken care to ensure that our wording is precise. About the case at iteration 0, we have checked thoroughly the implementation of the previous experiment, and found that the checkpoint termed with iteration number 0 is from the end of iteration 0, not before the iteration 0, which means that some optimizer steps have been performed at this point and that might explain why the pattern appears at that point. However, since this point can neither represent the case of random networks or sufficiently RL trained networks, we remove it from the plot to be not misleading.
> > >
> > > For the second paragraph of questions, we would like to respond to the last suggestion first:
> > >
> > > > I mean it would be a very compelling argument for using PIDM if you can show that it, whenever necessary, intelligently produces target joint positions different from the intended target joint position by the policy in order to make other components of the state match.
> > > >
> > >
> > > We first would like to draw your attention to some different implementation details of PIDM between pretraining stage and RL stage: during pretraining stage, PIDM’s input includes delta joint position; however during RL stage, its input does not include explicit delta joint position, but the output of intention encoder in the feature space. So technically it is hard to extract the deltas from that feature space in RL stage and perform suggested demonstration. Moreover, we would say that the idea of PIDM is not to balance between tracking different components of the state, instead, is to have pretrained dynamics knowledge before the launch of RL training.
> > >
> > > > … have you looked into how much the PIDM deviates from just adding the deltas to the current joint angles to produce target joint positions?
> > > >
> > >
> > > PIDM is designed to act as a solver of inverse dynamics during pretraining that maps the deltas to control signal. So the question translates into “how different is the inverse dynamics from an identity function?” Normally the (inverse) dynamics of physics simulation of robotic tasks is complex and highly non-linear and therefore discrepant from an identity function, and so will be the PIDM that trained to fit the highly non-linear mapping.
> > >
> > > > Also, how does your method compare to a vanilla MLP that produces delta joint positions instead of absolute joint positions?
> > > >
> > >
> > > We worry that the direct comparison between our method and a vanilla MLP that produces delta joint positions might not be well controlled, due to the coexistence of differences in both action space and whether pretraining is involved. Vanilla MLPs that produces delta joint positions are relatively less used in robot locomotion learning, and empirically would need more time to converge than absolute joint positions due to the different explorative behavior. The quality of policies learned with delta joint position as action space tend to be similar to those of absolute joint position in most common tasks.

---

> ### Author Response · Authors · 2025-11-21
>
> > **Questions**
> >
> > - What is the state and action space of the experiments with ANYmal?
> > - Why does the PIDM model require a history of proprioceptive states? Are joint velocities not included in x?
> > - In Figure 6, how many environment steps does each iteration amount to?
> > - How is the PIDM used in the critic?
>
> Thank you for the questions, and we will improve on the manuscript to make those points clearer.
>
> 1. Possible observation composition is listed in Table 6. The tasks either has or does not have access to exteroperception (Sec 5.1). For the detailed description of task-specific commands, please refer to the original works mentioned in Sec 5.1. As all tasks are formulated as POMDPs, a full state space includes aforementioned observation space, current noise levels, domain randomization parameters, and full terrain descriptions. The action space is joint position targets sent to the low-level ANYDrive controller.
> 2. The necessity of requiring a history of proprioception is mainly due to the absense of terrain information and contact state in proprioception, and the presence of noise and domain randomization techniques in the training process. Therefore, it would be inappropriate to fit the PIDM with only one single frame of current proprioceptive state, due to the fact that one proprioceptive state can actually correspond to various actual full states. In addition, there is a line of works [1,2] that suggest that proprioception history is informative and a number of useful values (e.g. foot height, contact probability, end effector force) can be estimated based on it.
> Joint velocities are included in $x$, but are subjected to a high level of noise, because with real hardware joint velocities are derived by differencing joint position readings.
> 3. We followed the choice by all previous works: 24 steps in 4096 environments in parrellel, i.e. 98,304 steps per iteration in total.
> 4. PIDM is used in the critic via almost identical architecture suggested in Fig. 3, with the only difference that the "Action Synthesizer" is replaced with a "Value Synthesizer" that outputs a scalar value estimation optimized with MSE loss.
>
> [1] Ji, Gwanghyeon, Juhyeok Mun, Hyeongjun Kim, and Jemin Hwangbo. "Concurrent training of a control policy and a state estimator for dynamic and robust legged locomotion." IEEE Robotics and Automation Letters 7, no. 2 (2022): 4630-4637.
>
> [2] T. Portela, G. B. Margolis, Y. Ji and P. Agrawal, "Learning Force Control for Legged Manipulation," 2024 IEEE International Conference on Robotics and Automation (ICRA), Yokohama, Japan, 2024, pp. 15366-15372

---

> > ### Comment · Reviewer_QTfi · 2025-11-24
> >
> > Thank you for these clarifications. Please consider revising the manuscript to make them available to readers.

---

### Official Review · Reviewer_L9yJ · 2025-11-05

**Soundness:** 3
**Presentation:** 3
**Contribution:** 2
**Rating:** 4
**Confidence:** 3

**Summary:**

This paper presents a method for pretraining neural network weights to warm-start actor-critic reinforcement learning (RL) in robotic tasks.
The core idea is to pretrain a Proprioceptive Inverse Dynamics Model (PIDM) using task-agnostic, exploration-based data.
This pretrained model is then integrated into the actor and critic networks of a PPO algorithm, providing a better initialization than random weights.
The method is evaluated on quadrupedal robot tasks in simulation, demonstrating improvements in final performance and sample efficiency compared to using randomly initialized networks.

**Strengths:**

1. A novel method of initializing the networks for RL training.
2. Clear presentation and thorough empirical analysis about the methods.
3. An interesting finding in section 5.2: it suggests that policy networks naturally learn a structure where earlier layers predict future states, and later layers invert this to compute actions. This reminds me of recent work in solving robot manipulation tasks by learning a world model to predict future observations (or states) first and predicting the actions from an inverse dynamics model.

**Weaknesses:**

1. While the method does not require expert demonstrations, the exploration-based data collection phase itself involves training a separate PPO policy. The computational cost and time required for this pretraining stage are non-trivial and are not quantitatively compared against the sample efficiency gains in the main RL tasks. A discussion on the net computational benefit would strengthen the paper.
2. The hypothesis (Section 4.1) and probing experiment (Section 5.2) suggest that policies first form an target state and then compute the inverse action. If this is the case, it is somewhat counter-intuitive to initialize the network with an inverse dynamics model (PIDM). A more aligned approach might be to initialize the earlier layers with a forward dynamics model and the later layers with the PIDM? The paper would benefit from a deeper discussion or an ablation on this point.
3. The paper motivates the method by mentioning its relevance to the sim-to-real gap (e.g., through noise and domain randomization). However, all experiments are conducted in simulation. For a robotics paper, real-world validation or a dedicated sim-to-real transfer experiment would significantly strengthen the claims about the practical effectiveness of this method.

**Questions:**

1. Why using $ \Delta x_{t+1}^* $ instead of $ x_{t+1}^* $? Does this design choice help bridge the distribution shift when switching the Delta Encoder to the task-specific Intention Encoder? Should using $x_{t+1}^*$ lead a smaller gap when replacing it with $o_t$?
2. From Figure 6, the PIDM with random initialization underperforms the vanilla MLP in some tasks (e.g., Locomotion) but performs similarly or better in others (e.g., Climb Up). What factors might explain this inconsistent performance?
3. The paper states that the method addresses the sim-to-real gap by incorporating noise and domain randomization. Beyond this, are there specific properties of the pretrained PIDM that make it particularly suitable for sim-to-real transfer?
4. This method reminds me of offline-to-online RL method [1]. How does this pretraining approach compare to alternative ways of leveraging the exploration data, such as:
1) Continue to train the actor and critic with normal RL training after the first phase of data collection using exploratory policy.
2) Using the data to fill the replay buffer for an off-policy RL algorithm.
3) Using the exploratory policy as a behavior policy for off-policy RL?

[1] Efficient Online Reinforcement Learning with Offline Data. Philip J. Ball, Laura Smith, Ilya Kostrikov, Sergey Levine. ICML 2023.

Minor
1. Please ensure all notations are clearly defined upon first use. For example, the definition of $ \Delta x_{t+1}^* $ is not clear. $\Delta x_{t+1}^* = x_{t+1}^*-x_{t}$, is this correct? Why using a * here?

---

> ### Author Response · Authors · 2025-11-21
>
> We thank you for your valuable review and feedbacks.
>
> > **Weakness 1** While the method does not require expert demonstrations, the exploration-based data collection phase itself involves training a separate PPO policy. The computational cost and time required for this pretraining stage are non-trivial and are not quantitatively compared against the sample efficiency gains in the main RL tasks. A discussion on the net computational benefit would strengthen the paper.
> >
>
> We agree that the exploration-based data collection entails certain computational cost as well. However, we present evidence that by running the data collection and pretraining for only ONE time for a single embodiment, we facilitate considerable sample efficiency improvements on various downstream tasks, and we reasonably suppose that this would also be helpful to more tasks that are yet to come, given that the pretrained PIDM can be effortlessly integrated to arbitrary downstream MDP design that includes proprioception in its observation. Therefore as a one-time weights initialization method, we believe that comparing its computational budget to an unspecified number of RL processes for training task-specific policies is technically difficult and may not lead to a meaningful or fair assessment. For more clarifications of our problem setup please see our response to Question 4.
>
> > **Weakness 2** The hypothesis (Section 4.1) and probing experiment (Section 5.2) suggest that policies first form an target state and then compute the inverse action. If this is the case, it is somewhat counter-intuitive to initialize the network with an inverse dynamics model (PIDM). A more aligned approach might be to initialize the earlier layers with a forward dynamics model and the later layers with the PIDM? The paper would benefit from a deeper discussion or an ablation on this point.
> >
>
> This is a good point of observation. We agree that an intuitive approach is to initialize the earlier layers with a forward dynamics model. However, a forward dynamics model not only requires the information about current state but also the action to take at timestep $t$. But the function of the entire policy network is to get the action, so either we do not have sufficient input due to the missing action before the forward process reaches the end of policy network, or we have already produced the action at the end of policy network and thus there is no need for a forward dynamics model. This complication of “chicken or egg which comes first" makes the idea of utilizing a forward dynamics model very twisted and hardly practical. Yet we think of this as an interesting direction to explore in future works, but not of significance to our current work.
>
> > **Weakness 3** The paper motivates the method by mentioning its relevance to the sim-to-real gap (e.g., through noise and domain randomization). However, all experiments are conducted in simulation. For a robotics paper, real-world validation or a dedicated sim-to-real transfer experiment would significantly strengthen the claims about the practical effectiveness of this method.
> >
>
> To put it more precisely, addressing sim-to-real transfer is not the main contribution or motivation of our method, but because we have successfully experimented with a wide range of task-specific RL processes in the presence of notable noise and domain randomization, the compatibility of our method to these techniques that commonly used to facilitate sim-to-real transfer has been tested, and thus we believe it is intrinsically more applicable for sim-to-real transfer than other works that have not considered sim-to-real transfer in their designs.
>
> Because 1) addressing sim-to-real is not our main contribution, and 2) limitation of the time frame, we will not deploy every mentioned controller on real hardware, but we will work on deploying the "locomotion" policy to demonstrate that it can transfer to real robot.

---

> ### Author Response · Authors · 2025-11-21
>
> > Questions 1 Why using $ \Delta x_{t+1}^* $ instead of $ x_{t+1}^* $? Does this design choice help bridge the distribution shift when switching the Delta Encoder to the task-specific Intention Encoder? Should using $x_{t+1}^*$ lead to a smaller gap when replacing it with $o_t$?
> >
>
> Using $ \Delta x_{t+1}^* $ comes from the simple fact that the reachable next proprioceptive state is mostly in the neighborhood of current proprioceptive state. Therefore $ \Delta x_{t+1}^* $ is obviously less correlated to $ x_t $ than $ x_{t+1}^* $. Thus we speculate that using the formulation of $ \Delta x_{t+1}^* $ would be at least similar, if not better than $x_{t+1}^* $ during pretraining. We have not tested if using $ \Delta x_{t+1}^* $ instead of $ x_{t+1}^* $ helps bridge the distribution shift when switching Delta Encoder to the task-specific Intention Encoder, but we suppose either way is actually feasible in practice.
>
> > Questions 2 From Figure 6, the PIDM with random initialization underperforms the vanilla MLP in some tasks (e.g., Locomotion) but performs similarly or better in others (e.g., Climb Up). What factors might explain this inconsistent performance?
> >
>
> The difference between PIDM with random initialization and vanilla MLP generally lies in these 2 facets: 1) PIDM (Random Init) has access to several more history frames of observation, while vanilla MLP has not; 2) PIDM (Random Init) is of larger size than vanilla MLP. Therefore, the variation of comparison can be dependent on whether single-timestep observation is enough for the application, and whether the capacity of vanilla MLP is already enough to learn this policy. In addition, it’s worth to note that we directly reuse all hyperparameter values from the well-tuned vanilla MLP baselines which might not be optimal for PIDM (Random Init). Further hyperparameter tuning might also help to improve the performance of PIDM (Random Init).
>
> > Questions 3 The paper states that the method addresses the sim-to-real gap by incorporating noise and domain randomization. Beyond this, are there specific properties of the pretrained PIDM that make it particularly suitable for sim-to-real transfer?
> >
>
> As our method 1) adds noise vectors to every batch of training data  during pretraining, and 2) allows noise and randomization implementations in vanilla MLP baselines to be kept, we believe it is intrinsically more suitable for sim-to-real transfer than works without these properties. Furthermore, we suspect that the pretraining might help the final policy to be more stable in response to out-of-distribution observation (observations that are out of the coverage of conducting the specific task), for that the PIDM is trained on a diverse exploration dataset beforehand. We leave the validation of this guess for future works.
>
> > Questions 4 This method reminds me of offline-to-online RL method [1]. How does this pretraining approach compare to alternative ways of leveraging the exploration data, such as
> >
> > 1. Continue to train the actor and critic with normal RL training after the first phase of data collection using exploratory policy.
> > 2. Using the data to fill the replay buffer for an off-policy RL algorithm.
> > 3. Using the exploratory policy as a behavior policy for off-policy RL?
> >
> > [1] Efficient Online Reinforcement Learning with Offline Data. Philip J. Ball, Laura Smith, Ilya Kostrikov, Sergey Levine. ICML 2023.
> >
>
> We would like to clarify that our work addresses a different problem compared to the majority of offline-to-online RL methods. We aim to provide a task-agnostic weights initialization for all possible downstream tasks of the specific embodiment. The unknownness and possible variation of downstream MDPs determine that it is impossible to include task-specific reward signal in the pretraining dataset. The method in [1] and the ideas listed above clearly require the task MDP is fully known and free to explore in advance, which is not available under the setup of our problem. Moreover, cross-task transfer is not demonstrated in [1] either.
>
> > Minor Please ensure all notations are clearly defined upon first use. For example, the definition of $ \Delta x_{t+1}^* $ is not clear. $\Delta x_{t+1}^* = x_{t+1}^*-x_{t}$, is this correct? Why using a * here?
> >
>
> That is correct. We simply use $x_{t+1}^* $ to emphasize that it denotes the *desired* next state. This indicates that states other than those that actually occurred can be used to query an action from the model. We will revise the manuscript to better clarify this notation.

---

> ### Comment · Reviewer_L9yJ · 2025-11-25
>
> Thank the authors for clarifying the questions.
> The tasks considered in this paper have clearly defined reward functions, right?
> If true, for question 4, it is easy to label the transitions/trajectories collected during exploration stage and treat it as a reply buffer for downstream tasks. So for the three variations I mentioned:
> 1. Variation 1 might fail considering the gap between the value function defined by the exploratory reward and the value function defined by the downsteam task reward.
> But this is another way to initialize the networks.
> 2. Variation 2 and 3 might be other options to use the exploratory data.
> 3. Combining 2 and 3 might lead to a better way to use the exploratory data?
>
> Considering the limited evaluations and ablation study in this paper, I encourage the authors to compare the proposed initialization method with other methods of initialization or other methods of using the exploratory data to improve the contribution of the paper.
> Moreover, more experiments or analysis on why this initialization helps can further strengthen the paper.
> For example, measuring the gap between the parameters of the networks at the initialization and at the convergence might be another way to define the effectiveness of the initialization.

---

> > ### Author Response · Authors · 2025-12-03
> >
> > We acknowledge that labeling the transitions/trajectories collected during exploration stage with task-specific reward functions can be another potentially interesting direction. However, in our perspectives, such relabeling is not easy and significant gaps between exploration and task-specific training need to be bridged, including:
> >
> > 1. Gap of observation space. As shown in Table 7, it is an often case that the observation space of task-specific training is different from that of exploration, more specifically in terms of the inclusion of task-specific command and more sensory information. Such change in observation would make naive “continue training” in idea 1 infeasible, necessitating architectural changes of the actor and critic networks to deal with such discrepancy. This also makes the reward relabeling not straightforward: missing parts of observation like commands and height scans need to be generated accordingly, and such regeneration can further require more information than the saved $(x_t, a_t)$ tuples. Then as a result, the offline dataset needs to save a complete history of all simulation details, including terrains, contact states and interaction forces to be able to provide the necessary values for all possible task-specific reward formulation, which is beyond the commonly-used trajectory collection.
> > 2. With regard to Idea 2 and 3, a practical consideration is generally off-policy methods have rarely been demonstrated to succeed in highly dynamic locomotion tasks, so the quality of such baseline method is questionable. Or, if the relabeled transitions are to be added into the set of on-policy data during PPO update, then extra work would be necessary to handle the distribution mismatch incurred by using off-policy data in policy update of on-policy methods.
> >
> > Considering all above aspects, we believe that though the mentioned ideas can be potentially interesting, they remain potential ideas that would require extensive efforts to develop rather than off-the-shelf baselines in our application. If implemented, one such method would merit a standalone publication of itself, and is therefore out of the scope of this work.
> >
> > Regarding the suggested experiment “measuring the gap between the parameters of the networks at the initialization and at the convergence”, we would like to highlight that we did do a very similar study of measuring the lengths of optimizer step in the parameter space, presented in Section 5.6 and A.9. We believe that compared to simply measuring the gap between the parameters of the networks at the initialization and at the convergence, plotting per-iteration update magnitude is more informative, as it reveals how the parameters evolve throughout training and offers insight into the geometry of the loss landscape. The experiments show that with our initialization, the update magnitudes are smaller in everywhere inside the actor and critic networks, verifying the effectiveness of our proposed initialization.

---

### Author Response · Authors · 2025-12-03
**General Response**

Dear Area Chair and Reviewers,

We are delighted and would like to thank the reviewers for their acknowledgements:

- Address important challenge (Reviewer QTfi); interesting and timely problem setting (Reviewer J3PS).
- Novel method (Reviewer L9yJ); consistent performance gains (Reviewer RrdJ, Reviewer HATj);
- Good compatibility (Reviewer QTfi); generally applicable (Reviewer J3PS).
- Thorough empirical analysis (Reviewer L9yJ); well chosen evaluations (Reviewer J3PS); broad and well-controlled experiments, convincing ablations (Reviewer HATj); insightful ablations and analyses (Reviewer RrdJ).

We also thank the reviewers for all the constructive feedbacks. We summarize here the improvements we have made to our submission based on feedbacks from reviewers:

1. **Clarification about “Insufficient baselines” (addressing Reviewer QTfi, RrdJ, HATj, J3PS)** We would like to clarify that our work addresses a different problem compared to the model-based and offline-to-online RL methods referenced by reviewers: we aim to provide task-agnostic weights initialization for all possible downstream tasks of the specific embodiment. Most notably, due to the unknownness and possible variation of downstream MDPs, the pretraining dataset does not include any task-specific reward signals. The characteristics of this formulation have made the comparison between our proposed method and many suggested baselines inappropriate, which we have detailed in our replies. We have improved *Introduction* and *Related Works* chapter of the manuscript to better clarify the positioning of our method with respect to other works.
2. **Added experiments with more embodiments (addressing Reviewer QTfi, HATj, J3PS)** We have added experiments with Unitree Go1 (quadruped) and Unitree G1 (humanoid) to main experiments results in Section 5.4, where the effectiveness of the proposed method is further verified and strengthened.
3. **Addressing concern about accuracy and what is transferred (addressing Reviewer RrdJ, HATj)** We have added the ablation that shows how different PIDM’s error levels affect improvements observed during RL training to Appendix A.8. The results showcase a trend that generally models of higher accuracy tend to provide larger gains in RL stage.
4. **Improved dynamics knowledge probing experiment (addressing Reviewer QTfi)** We have improved the formulation of experiment “Dynamics knowledge probing in vanilla RL policy networks”. Specifically:
    1. To exclude the possibility that the worse prediction based on representations from deeper layers is due to the loss of information about current state, we append the raw observation into the input of the lightweight prediction MLP to always provide the same amount of information about the current state.
    2. We scale the experiment to 2 tasks, each task with 5 random seeds, and visualize the standard deviation of the data points and the magnitude of ground truth change in joint angles in the plot.

    The updated experiment continues to show the same pattern of error levels of representations, presented in Figure 4b and Appendix A.2.

5. **Added sim-to-real demonstration (addressing Reviewer L9yJ, QTfi).** We have deployed the policy for one task “locomotion” on real hardware to showcase the sim-to-real transfer capability of our method (Appendix A.10). Video is attached in the [supplementary materials](https://openreview.net/attachment?id=2vBAf3J0uD&name=supplementary_material). Due to time limitations, we were not able to complete deployments of all 9 skills in the paper, but we can reasonably suppose that these policies would transfer too, since sim-to-real transfer of these tasks have all been tested in the original works, and the training configurations have not been fundamentally altered.
6. **Improved overall clarity of the manuscript and the codebase (addressing Reviewer QTfi, RrdJ).**
    1. We rephrase the title and related statements to state more precisely that this method is intended specifically for robot locomotion.
    2. We have added more implementation details including those inquired by Reviewer QTfi.
    3. We have added a readme file in the code supplements to enable easier navigation within the codebase.

---

### Meta-Review · Area_Chair_ED9q · 2026-01-04

**Summary:**

Reviewers agree the paper tackles a relevant problem—task-agnostic pretraining for warm-starting PPO-style actor–critic learning in robot locomotion—and note consistent gains plus thoughtful ablations/analyses. The key concerns were (i) insufficient/unclear baselines and controls beyond random init (e.g., requests for adapted pretraining/weight-sharing baselines and a control that “pretrains the policy itself” on the same exploration data), making it hard to isolate what is specific to the PIDM decomposition versus generic initialization; (ii) practical cost/benefit of the exploration+pretraining stage; (iii) transfer/generality (initially limited embodiment coverage; questions about manipulators/bipeds); (iv) what is actually transferred, given reported PIDM prediction errors (~40–50%); and (v) sim-to-real evidence (initially all-sim). The authors provided detailed rebuttals, added experiments on additional embodiments (Unitree Go1, Unitree G1), added an ablation relating PIDM accuracy to RL gains, improved the probing study, and included a sim-to-real locomotion deployment. However they did not add strong baselines as requested by the reviewers. While the authors argue that unsupervised or offline-to-online RL methods are not directly comparable due to differing objectives, the lack of any empirical comparison to alternative ways of leveraging the same exploration data (e.g., unsupervised representation learning or reward-free RL pretraining) remains a significant limitation. Without such comparisons, it is difficult to isolate whether the observed gains stem from the specific inverse-dynamics-based initialization or from more general benefits of pretraining on exploration data.

Separately, the authors flagged two reviews as potentially fully AI-generated; regardless, the critiques in those reviews were concrete and were addressed on technical grounds in the rebuttal.

**Reviewer Concerns:**

Addressed concerns:

- Generality across embodiments: The authors added experiments on additional robot embodiments (Unitree Go1 and Unitree G1), strengthening the claim that the proposed pretraining strategy captures embodiment-specific knowledge transferable across multiple tasks.

- Representation transfer: Additional ablations relating PIDM prediction accuracy to downstream RL gains help clarify that higher-quality inverse-dynamics representations correlate with improved learning performance.

- Sim-to-real transfer: A real-robot deployment for the locomotion task was added, partially addressing concerns that the evaluation was entirely simulation-based.

- Clarity: The authors improved the explanation of the method’s scope and clarified how it differs from offline-to-online RL and skill-based pretraining approaches.

Outstanding concerns:

- Lack of strong baselines: Comparisons remain largely limited to random initialization; alternative ways of using the same exploration data (e.g., pretraining policy networks directly or using the data for offline/off-policy RL), or the unsupervised RL that shares similar objectives to the proposed work were not empirically evaluated.

- Cost–benefit tradeoff: The computational cost of the exploration-based data collection and pretraining stage is still not quantitatively compared against the downstream gains.

- Methodological novelty: Even after rebuttal it remains unclear if the method offers a sufficiently novel insight beyond a well-engineered inverse-dynamics-based initialization strategy.

**Reviewer Scores:**

- Reviewer L9yJ: This reviewer engaged in discussion and acknowledged the authors’ clarifications and added experiments (additional embodiments, improved probing, sim-to-real demonstration). A modest increase in score is plausible, though remaining baseline and attribution concerns would not have led to a strong accept.

- Reviewer QTfi: Although this reviewer engaged in discussion, their main concern regarding insufficient comparisons to alternative ways of leveraging exploration data (e.g., unsupervised or reward-free RL) was addressed conceptually rather than empirically. I expect no score change after rebuttal.

- Reviewer RrdJ: Concerns about attribution and whether the observed gains are specific to the proposed pretraining strategy remain insufficiently resolved. Those concerns would lead to no change in score.

- Reviewer HATj:  While the reviewer acknowledged the thorough empirical evaluation, doubts about the level of novelty and necessity of the proposed approach persist after rebuttal. Those concerns would lead to no change in score.

- Reviewer J3PS: This reviewer viewed the problem setting positively but did not indicate that the rebuttal fundamentally altered concerns related to positioning and baseline adequacy. Those concerns would lead to no change in score.

---

### Decision · Program_Chairs · 2026-01-26

Reject